# Impossibility results for fair representations

## Abstract

With the growing awareness to fairness in machine learning and the realization of the central role that data representation has in data processing tasks, there is an obvious interest in notions of fair data representations. We provide a formal framework for examining the fairness of data representations through the lens of their effect on decisions (mainly classification) made based on data represented that way. Using that framework, we prove that several desiderata for fair representations cannot be achieved. While some of our conclusions are intuitive, we formulate (and prove) crisp statements of such impossibilities, often contrasting impressions conveyed by many recent works on fair representations.

## 1 Introduction

Automated decision making has become more and more successful over the last few decades and has therefore been used in an increasing number of domains, either as stand alone, or to support human decision makers. This includes many sensitive domains which significantly impact people's livelihoods, such as loan applications, university admissions, recidivism predictions, or insurance rate settings. It has been found that many such decision tools have, often unintentionally, biases against minority groups, and therefore lead to discrimination. In response to these concerns, the machine learning research community has been devoting effort to developing clear notions of fair decision making, and coming up with algorithms for implementing fair machine learning.

A common approach to address the important issue of fair algorithmic decision making is through *fair data representation*. The idea is that some regulator or a responsible data curator transforms collected data to a format (or *representation*), that can then be used for solving downstream classification tasks providing guarantees of fairness. This approach was proposed by the seminal paper of Zemel et al. [15]. In their words: "our intermediate representation can be used for other classification tasks (i.e., transfer learning is possible)"... "We further posit that such an intermediate representation is fundamental to progress in fairness in classification, since it is composable and not ad hoc; once such a representation is established, it can be used in a blackbox fashion to turn any classification algorithm into a fair classifier, by simply applying the classifer to the sanitized representation of the data". Many followup papers aim to realize this paradigm, solving technical and algorithmic issues [10, 6, 11, 14, 3] (to mention just a few). The main contribution of this paper is showing that, basically, *it is impossible to achieve this goal*. Namely, no data representation can guarantee that for every classification task a classifier trained on data under the given representation will be fair for that task. This impossibility applies even if one restricts the downstream tasks in question to share the same labeling rule, or for fairness notions like Odds Equality, to share the same marginal data distribution with the data on which the representation was trained. Our results answer negatively the main two questions posed in the discussion section of Creager et al. [3].

While many papers in this domain propose algorithmic solutions to fairness related issues, the main contributions of this paper are conceptual. We believe that, to a much larger extent than many other

facets of machine learning, fundamental concepts of fairness in machine learning require better understanding. Some basic questions are still far from being satisfactorily elucidated; What should be considered fair decision making? (various mutually incompatible notions have been proposed, but how to pick between them for a given real life application is far from being clarified). What is a fair data representation? To what extent should accuracy or other practical utilities be compromised for achieving fairness goals? and so on. The answers to these questions are not generic. They vary with the principles and the goals guiding the agents involved (decision makers, subjects of such a decision, policy regulators, etc.), as well as with what can be assumed regarding the underlying learning setup. We view these as the primary issues facing the field, deserving explicit research attention (in addition to the more commonly discussed algorithmic and optimization aspects). This is a theoretical work, our discussion is grounded in definitions and proofs rather than heuristics and experimental results.

## 1.1 What is *fair representation*?

The term 'fair data representation' encompasses a wide range of different meanings. When word embeddings results in smaller distance between the vectors representing 'woman' and 'nurse' relative to the distance between the representations of 'woman' and 'doctor' and the other way around for 'man', is it an indication of bias in the *representation* or is it just a faithful reflection of a bias in society? Rather than delving into such issues, we discuss an arguably more concrete facet of data representation; We examine representation fairness from the perspective of its effect on the fairness of classification rules that agents using data represented that way may come up with. Such a view takes into consideration two setup characteristics:

**The objective of the agent using the data** We distinguish three types of classification prediction agents (formal definitions of these aspects of fairness are provided in section 3.2):

*Malicious* - driven by a bias against a group of subjects. To protect against such an agent, a fair representation (or feature set) should be such that *every* classifier based on data represented that way is fair. This is apparently the most common approach to fair representations in the literature e.g., [15, 10].

*Accuracy Driven* - focusing on traditional measures of learning efficiency, ignoring fairness considerations. A representation is accuracy-driven fair if every loss minimizing classifier based on that representation is fair.

*Fairness Driven* - aiming to find a decision rule that is fair while maintaining meaningful accuracy. A representation is fairness-driven fair if there exists a loss minimizing (or an approximate minimizer) classifier based on that representation is fair.

**The notion of group fairness applied to the classification decisions** The wide range of group fairness notions (for classification) can be taxonomized along several dimensions: Does the notion depend on the ground truth classification or only on the agents decision (like demographic parity)? Is perfectly accurate decision (matching the ground truth classification) always considered fair (like in odds equality)? Does the fairness notion depend on unobservable features (like intention or causality)? In this work we focus on fairness notions that are ground-truth-dependent, view the ground truth classification as fair and depend only on observable features. The decision which notion of fairness one wishes to abide by depends on societal goals and may vary from one task to another and is outside the scope of this paper. Just the same, let us briefly explain why the requirements listed above are natural in many situations.

*The dependence on the ground truth classification* is almost inevitable from a utilitarian perspective - taking into account the probability that a student succeed or fail when making acceptance decisions should not be considered unfair. Put more formally, whenever there is any correlation between membership and the ground truth classification, any classifier that is fair w.r.t. a notion that ignored the ground truth (like demographic parity) is bound to suffer prediction error proportional to that correlation.

*Viewing perfectly accurate decisions as fair* can be viewed as a distinction between notions that do or do not try to inflict affirmative action. It makes a lot of sense in tasks like conviction in a crime - if you convict all criminals and no one else, you should not be accused on unfairness.

*Relying only on observable features* fosters objectivity and allows scrutiny of the decisions made. Our running example of such a notion is odds equality [8], however our results

hold as well for other common notions of fairness that meet the above conditions (like Calibrations Within Groups [9]).

## 1.2 Our results

We prove the following inherent limitations of notions of fair representations (under the above taxonomy):

1. *The impossibility to be task-independent*. There is a host of literature proposing methods of coming up with data representation that guarantees the fairness of classifier based on that representation (e.g., [18, 3, 10, 12]). We elaborate on these works in our Previous Work section. Contrasting the impression conveyed by many such papers, we show that the ability to guarantee multi-task fairness is inherently limited. Much of that work addresses Demographic parity (DP). We prove that if two tasks have different marginal data distributions (that is, the distribution of unlabeled instances) and different success rates of the protected group, then no representation can guarantee that any non-trivial classifier trained on it satisfies DP for both. We show that the only classifiers that are guaranteed to satisfy any significant level of DP fairness w.r.t. all marginal distributions are the redundant constant functions. From a practical point of view, since DP fairness of some decision (say, acceptance to some university program) requires the ratio of positive decisions between groups to match the ratio of applicants from those groups, a representation that guarantees DP fairness cannot be a priory constructed - it must have access to the distribution of groups among applicants for that specific program. Furthermore, we prove that for every fixed marginal data distribution, if two ground truth classifications differ with non-zero probability over it, *there can be no data representation that enjoys Odds Equality fairness and accuracy with respect to both tasks over that shared marginal distribution* (except for the redundant case where the success rates of both groups are equal for both tasks). These results answer negatively the main two open problems posed in the Discussion section of [3].

2. *The impossibility to evaluate the fairness contribution of a given feature devoid of the other features used* (again, for each agent objective and several common group fairness notions).

3. *The inherent dependence of the effect on fairness of adding/deleting a feature on the type of agent using the representation* (on top of the above mentioned dependence on other features), even when the feature in question does not correlate with membership in the protected group.

(These come on top of the obvious dependence on the notion of fair classification sought).

**Concerning potential negative societal impact:** We cannot foresee any potential negative societal impact of our work. The main message of this paper is a cautionary statement. We alert potential users that approaches based on task independent fair representations cannot guarantee the fairness of arbitrary predictors based on them. As such, we are only guarding against potential negative impact of previously published work.

Our paper is organized as follows: Section 2 gives an overview of the related work. Section 3 introduces our setup including our taxonomy for fair representations. Section 4 contains our main results on the impossibility of generic fairness of a representation. Section 5 addressed the impossibility of defining the fairness effect of a single feature without considering the other components of a representation. Section 6 briefly shows the impossibility of having fair representations w.r.t. Predictive Rate Parity. Section 7 is our concluding remarks.

We defer proofs to the appendix.

## 2 Related Work

Since our paper goes against messages conveyed by many previous papers, we wish to address in detail more related works than space here allows. We therefore provide a more elaborate section on previous work in the supplementary material.

Much of the recent work on fair representation for learning classifiers focuses on algorithms. (and demonstrating the viability of those algorithms though experimental results) [15, 10, 17, 1, 16]. As explained before, our focus is different. We discuss what should be considered fair representation in

that context, what is the scope of such notions and what are the inherent limitations of defining such representations.

Almost all the work on fair representations focuses on the demographic parity (DP) notion of fairness [6, 10, 15, 14]. Not having to take ground truth into account makes this notion independent of the classification task carrying both advantages and limitations. However, any positive result in these papers assumes that the marginal data distribution is available to the designer of the fair representation. Such an assumption severely restricts the applicability of such representations. To achieve DP fairness, a classifier has to induce success ratio between the two groups that match the ratio between these groups in the input data. However, that ratio, say a set of applicants for a bank loan or to some university program varies from one application to another and cannot be determined a priori. Our results on this inherent limitation of fair representation for DP (see section 4) do not seem to have been stated before.

When the data marginal distribution is fixed, and available to the designer of a representation, DP fairness is possible. However, in such a setup, we show that fairness with respect to notions of fairness that do rely on the correct ground truth, such as equalized odds (EO) [8], cannot be guaranteed for arbitrary tasks (see Section 4). This fact also has not been explicitly stated (and proved) before, although it seems that some of the previous work worried about it. Instead, previous work either focus only on DP fairness, or, when it comes to discuss other notions of fairness, the algorithms that design the representations are assumed to have access to task specific labeled data (e.g. [16, 2, 14, 5], which defies the goal of having a fixed representation that guarantees fairness for many tasks.

The effect of the motivation of the decision maker using the representation on the fairness of the resulting decision rule has been considered by Madras et al. [10] and Zhang et al. [16]. These papers identify two motivations. The first is malicious, which is the intent to discriminate without regard for accuracy. The second is accuracy-driven, which is the intent to maximize accuracy. We address these effects as part of our taxonomy of notions of fair representations. Additionally, we discuss *fairness-driven agents* that aim to achieve fairness while maintaining some level of accuracy.

A natural question that arises in this context is about the inherent trade-offs between fairness and accuracy. When the notion of fairness is demographic parity, such trade-offs are clearly expected - they surface whenever there exists correlation between membership in the protected group and the ground truth classification. Zhao et al. [18] and Mcnamara et al. [11] analyze such scenarios and demonstrate situations in which there exists a more accurate and more fair classifier based on an original representation than any classifier built using a learnt representation.

The question of feature deletion has also been considered in real world examples, such as in the "ban the box" policy which disallowed employers using criminal history in hiring decisions [4]. The effect of allowing or disallowing features on fairness has been studied before, for example in Grgic-Hlaca et al. [7]. However in previous works, the effect of a feature on fairness, has been discussed in isolation. In contrast, we show that fairness of a feature should not be considered in isolation, but should also take into account the remaining features available.

## 3   Formal Setup

We consider a binary classification problem with label set $\{0, 1\}$ over a domain $X$ of instances we wish to classify, e.g. individuals applying for a loan. We assume the task to be given by some distribution $P$ over $X \times \{0, 1\}$ from which instances are sampled i.i.d. We denote the ground-truth labeling rule as $t : X \to [0, 1]$. We will think of the label 1 as denoting 'qualified' and the label 0 as 'unqualified' and $t(x) = P[y = 1|x]$. For concreteness, we focus here on the case of deterministic labeling (that is $t : X \to \{0, 1\}$). Most of our discussion can readily be extended to the probabilistic labeling case. In a slight abuse of notation we will sometimes use $t(w)$ to indicate the label coordinate of an instance $w \in X \times \{0, 1\}$

A data representation is determined by a mapping $F : X \to Z$, for some set $Z$, and the learner only sees $F(x)$ for any instance $x$ (both in the training and the test/decision stages).We denote the hypothesis class of all feature based decision rules as $\mathcal{H}_F = \{h : Z \to \{0, 1\}\}$. As a loss function

we consider a weighted sum of false positives and false negatives, i.e.

$$l^\alpha(h, x, y) = \begin{cases} \alpha, & if \ h(x) = 0, y = 1 \\ 1 - \alpha, & if \ h(x) = 1, y = 0 \\ 0, & otherwise \end{cases}$$

for some weight $\alpha \in (0, 1)$. We denote the true risk with respect to this loss as $L_P^\alpha$ and the empirical risk as $L_S^\alpha$.

### 3.1 Notions of group fairness

For our fairness analysis we assume the population $X$ to be partitioned into two subpopulation $A$ and $D$ (namely, we restrict our discussion the case of one binary protected attribute). We sometimes use a function notation $G : X \to \{A, D\}$ to indicate the group-membership of an instance. Of course in reality there are often many protected attributes with more than two values. However, as our goal is to show limitations and impossibility results for fair representation learning, it suffices to only consider one binary protected attribute – the same impossibilities readily follow for the more complex settings.

We now define two widely used notions of group-fairness that we will refer to throughout the paper, namely, equalized odds and demographic parity. In the following we will denote with $X_{g,l}$ the subset of $X$ with label $l$ and group membership $g$, i.e. $X_{g,l} = X \cap t^{-1}(l) \cap G^{-1}(g)$.

**Definition 1 (Group fairness; Equalized odds)** *The notion of group-fairness we will focus on in this paper is the ground-truth-dependent notion of odds equality as introduced by [8].*

*A classifier $h$ is considered fair w.r.t. to odds equality ($L^{EO}$) and a distribution $P$ if for $x \sim P$ we have the statistical independence $h(x) \perp\!\!\!\perp G(x)|t(x)$. For $g \in \{A, D\}$ let the false positive rate and the false negative rate be defined as $FPR_g(h, t, P) = \mathbb{P}_{x \sim P}[h(x) = 1|x \in X_{g,0}]$ and $FNR_g(h, t, P) = \mathbb{P}_{x \sim P}[h(x) = 0|x \in X_{g,1}]$ respectively. The EO unfairness is given then by the sum of differences in false positive rate and false negative rate between groups:*

$$L_P^{EO}(h) = \frac{1}{2}|FNR_A - FNR_D| + \frac{1}{2}|FPR_A - FPR_D|.$$

*If we say a classifier is fair, without referring to any particular group-fairness notion, we mean fairness w.r.t. equalized odds.*

**Definition 2 (Demographic parity)** *A classifier $h$ is considered fair w.r.t. to demographic parity ($L^{DP}$) and a distribution $P$ if $h(x) \perp\!\!\!\perp G(x)$. The respective unfairness is given by difference in positive classification rates between groups*
$$L_P^{DP}(h) = |\mathbb{P}_{x \sim P}[h(x) = 1|G(x) = A] - \mathbb{P}_{x \sim P}[h(x) = 1|G(x) = D]|.$$

### 3.2 The role of the agent's objective

We will phrase our definitions of representation fairness in terms of a general group fairness notion $L^{fair}$ with unfairness measure $L_P^{fair}$.

We start by considering a *malicious decision maker* who tries to actively discriminate against one group. To protect against this kind of decision maker, we need to give a guarantee such that based on the feature set it is not possible to discriminate against one group. This corresponds to the notion of adversarial fairness.

**Definition 3 (Adversarial fairness)** *A representation $F$ is considered to be* adversarial *fair w.r.t. the distribution $P$ and group fairness objective $L^{fair}$, if every classifier $h \in \mathcal{H}_F$ is group-fair. We define the adversarial unfairness of a representation $F$ by $U_{adv}(F) = \max_{h \in \mathcal{H}_F} L_P^{fair}(h)$.*

Furthermore, we consider an *accuracy-driven decision maker*, who aims to label instances correctly and is agnostic about fairness. For this kind of decision maker, we only need to make sure that optimizing for correct classification results in a fair classifier. The following definition ensures that the Bayes optimal classifier for a representation is fair.

**Definition 4 (Accuracy-driven fairness)** *A representation $F$ is considered to be* accuracy-driven *fair w.r.t. the fairness objective $L^{fair}$ and distribution $P$, if for every threshold $\alpha \in (0, 1)$, every classi-*

*fier $h \in \mathcal{H}_F$ with $L_P^\alpha(h) = \min_{h \in \mathcal{H}_F} L_P^\alpha(h)$ is group-fair. The accuracy-driven unfairness for a par-*
*ticular threshold parameter $\alpha$ is given by $U_{acc}^\alpha(\mathcal{F}) = \max\{L_P^{fair}(h) : h \in \arg\min_{h \in \mathcal{H}_F} L_P^\alpha(h)\}$.*
*The general accuracy-driven unfairness is given by $U_{acc}(\mathcal{F}) = \max_{\alpha \in [0,1]} U_{acc}^\alpha(\mathcal{F})$.*

We note that in cases where the decision maker does not have access to the distribution $P$, but only to a labelled sample, this requirement is might not sufficient for guaranteeing that an accuracy-driven decision maker arrives at a fair decision. In the Appendix we propose another fairness notion ($\lambda$-robustness) that formalizes the desired fairness guarantee for this scenario.

Lastly, we also consider a *fairness-driven decision maker* who actively tries to find a fair and accurate decision rule, while maintaining some accuracy guarantees. For such a decision maker a representation should allow for fair and accurate decision rules. If a representation fulfills this requirement, we call it fairness-enabling.

**Definition 5 (($\epsilon, \eta$)-fairness-enabling representation)** *A representation $F$ is considered to be ($\epsilon, \eta$)-fairness-enabling w.r.t. a fairness objective $L^{fair}$, if there exists a classifier $h \in \mathcal{H}_F$ that such that $L_P^\alpha(h) \leq \epsilon$ and $L_P^{fair}(h) \leq \eta$.*

Our discussion focuses primarily on the case of malicious and indifferent decision makers. These notions of fair representation can be defined with respect to any group-fairness notion. In our paper we will mainly focus on the equalized odds notion of fairness [8]. We also note that all the above definitions can be given with respect to a fixed model $\mathcal{H}$ in a continuous space.

# 4 Can there be a generic fair representation?

We address the existence of a multi-task fair representation. We prove that for the adversarial agent scenario (which is the setup that most fairness representation previous work is concerned with), **it is impossible to have generic non-trivial fair representations** - no useful representation can guarantee fairness for all "downstream" classification that are based on that representation (even if the ground truth classification remains unchanged and only the marginal may change between tasks).

We start by considering scenarios in which only the marginals shift between two tasks, e.g. two openings for different jobs, requiring similar skills, for which different pools of people would apply. Such a distribution shift can likely affect one group more than another and would thus affect the classification rates of both groups differently. We show that we cannot guarantee fairness of a fixed data presentation for general shifts of this kind, even for the simplest case of demographic parity.

**Claim: 1** *Pick any domain set $X$ and any partition of $X$ into non-empty subsets $A, D$. For every non-constant function $f : X \to \{0, 1\}$ there exists a probability distribution $P$ over $X$ such that $f$ is arbitrarily DP-unfair w.r.t. $P$ (say, $L_P^{DP}(h) > 0.9$).*

In particular, when a shift in marginal occurs between tasks, fairness for previous tasks does not imply a fairness guarantee for a new task.

**Proof:** *If $f$ is constant on any of the groups $A$ or $D$ then, since $f$ is not a constant over $X$ there is are points in the other group on which $f$ has the opposite value. Let $P$ assigns probability $0.5$ to the group on which $f$ is constant and probability $0.5$ to the set of points to which $f$ assigns the other value. Clearly $f$ fails DP w.r.t. this $P$. Otherwise, both values are assigned in both groups, so let $P$ assign probability $0.5$ to $\{x \in A : f(x) = 0\}$ and probability $0.5$ to $\{x \in D : f(x) = 1\}$. Clearly, $f$ fails DP w.r.t. this $P$.*

**Corollary 1** *No data representation can guarantee the DP fairness of any non-trivial classifier w.r.t. all possible data generating distributions (over any fixed domain set with any fixed partition into non-empty groups). That is, any non-constant representation F, cannot be adversarially fair with respect to $L^{DP}$ and any arbitrary task $P$.*

**Claim: 2** *Pick any domain set $X$ and any partition of $X$ into non-empty subsets $A, D$. For every non-constant function $f : X \to \{0, 1\}$ and every classifier $h : X \to \{0, 1\}$ such that $h \neq f$ there exists a probability distribution $P$ over $X$ such that $h$ is arbitrarily EO-unfair w.r.t. $P, f$, say $L_{P,f}^{EO} > 0.9$.*

**Corollary 2** *No data representation can guarantee EO fairness of any non-constant predictor based on that representation for all "downstream" classification learning tasks. That is, any non-constant representation F, cannot be adversarially fair with respect to $L^{EO}$ and any arbitrary task P. This holds even if one restricts the claim to tasks sharing a fixed marginal data distribution.*

We will now look at a slightly more restricted setting and analyse the case of multi-task learning, where instead of asking for a representation that is fair for every task, we only consider fairness with respect to a fixed (finite) set of tasks that we want to learn. We find that for the adversarial case, even this less ambitious goal is not achievable for generic tasks and the equalized odds notion of fairness.

We say a distribution $P$ has *equal success rates* if $\frac{P(X_{A,1})}{P(A)} = \frac{P(X_{D,1})}{P(D)}$.

**Lemma 1** *Let $P_1$ and $P_2$ be the distributions defining two different tasks with the same marginal $P_X = P_{1,X} = P_{2,X}$ such that at least one of the tasks does not have equal success rates. Let $h_1, h_2 : X \to \{0, 1\}$ be such that $L_{P_1}(h_1) = L_{P_2}(h_2) = 0$, and assume that tasks are non-negligibly different (namely, $L_{P_1}(h_2) \neq 0$). Then, it cannot be the case that both $h_1$ and $h_2$ are EO fair w.r.t. both $P_1$ and $P_2$.*

The proof (in the appendix) has a similar flavour as the proof of incompetability of different fairness notions of [9].

**Theorem 1** *There can be no data representation F such that for some $P_1, P_2$ as above, the following criteria simultaneously hold:*

1. *$\mathcal{F}$ is adversarially fair w.r.t. $P_1$ and EO*

2. *$\mathcal{F}$ is adversarially fair w.r.t. $P_2$ and EO*

3. *$\mathcal{F}$ allows for perfect accuracy w.r.t. to $P_1$ and $P_2$, i.e. there are $h_1, h_2$ both expressible over the representation F, such that $L_{P_1}(h_1) = L_{P_2}(h_2) = 0$.*

This result follows directly from Lemma 1. Therefore, if the goal is to prevent discrimination from a possibly adversarial decision maker, while also enabling accurate prediction, each task requires its task-specific feature representation.

# 5 Fairness of a feature set vs. fairness of a feature

In this section we discuss feature deletion and its impact on the fairness of a representation. For this we assume our representation $F$ to consist of finitely many features $f_i : X \to Y_i$ i.e. for every $x \in X : F(x) = (f_1(x), \dots, f_n(x))$ and $Z = Y_1 \times \dots \times Y_n$. We limit our discussion to cases where all $Y_i$ are finite. While this assumption facilitates our analysis, we do not expect our results to be different in the cases of continuous features. We will denote the set of features as $F = \{f_1, \dots, f_n\}$ and will denote by $U_{adv}(\mathcal{F})$ and $U_{acc}^{\alpha}(\mathcal{F})$ the adversarial and accuracy-driven fairness of the representation induced by the feature set $\mathcal{F}$ respectively. We show that it is in general not possible to determine the effect a single feature has on the fairness of a representation without considering the full representation. This is the case even if our considered feature is not correlated with the protected attribute.

## 5.1 Opposing effects of a feature for accuracy-driven fairness of a representation

We start our discussion with accuracy-driven fairness w.r.t. equalized odds. In this case we show that the deletion of a feature $f$ can lead to an increase in accuracy-driven unfairness for some set of other given features $\mathcal{F}$ and that the deletion of the *same* feature $f$ can lead to a decrease in accuracy-driven unfairness for another set of other available features $\mathcal{F}'$. This implies that the fairness of the feature $f$ cannot be evaluated without context. We show that this phenomena holds for a general class of features that satisfy some non-triviality properties (That on the one hand do not reveal too much information about group membership and labels (non-committing), and on the other hand does not reveal identity when label and group information is given ($k$-anonymity [13])). The exact definitions of these properties can be found in the appendix.

**Theorem 2** *(Context-relevance for fairness of features) For every* 6*-anonymous non-committing feature* $f$*, there exists a probability function* $P$ *over* $X$ *and feature sets* $\mathcal{F}$ *and* $\mathcal{F}'$ *such that:*

- *The accuracy-driven fairness w.r.t* $L^{EO}$*,* $P$ *and* $\alpha = 0.5$ *of* $\mathcal{F} \cup \{f\}$ *is greater than that of* $\mathcal{F}$*, i.e.*
$$U_{acc}^{\alpha}(\mathcal{F} \cup \{f\}) < U_{acc}^{\alpha}(\mathcal{F})$$
  *Thus, deleting* $f$ *in this context will increase unfairness.*

- *The accuracy-driven fairness w.r.t* $L^{EO}$*,* $P$ *and* $\alpha = 0.5$ *of* $\mathcal{F}' \cup \{f\}$ *is less than that of* $\mathcal{F}'$*, i.e.*
$$U_{acc}^{\alpha}(\mathcal{F}' \cup \{f\}) > U_{acc}^{\alpha}(\mathcal{F}')$$
  *Thus, deleting* $f$ *in this context will decrease unfairness.*

This phenomenon can happen even if $\{f\}$ is adversarially fair w.r.t. to $P$ and equalized odds.

## 5.2 The fairness of a feature for different notions of fairness

We will now briefly discuss the effect of a single feature on fairness for the cases of a malicious or a fairness-driven decision makers. In contrast to the accuracy-driven case, adding features has a monotone effect on the fairness of a fairness-driven and the malicious decision maker. As Theorem 3, adding any feature in the malicious case, will only give the decision maker more information and thus give the decision maker more chances of discrimination. Similarly in the fairness driven case, any feature will only give the decision maker another option for fair decision making (Theorem 4). However, the quantitative effect of adding a feature on the unfairness can still range from having no effect to achieving perfect fairness/unfairness for both the fairness-driven and the malicious case. As in the accuracy-driven case, we will show (Theorem 4 and Theorem 3) that it is impossible to evaluate the quantitative effect of a feature on the fairness of a representation without considering the context of other available features.

**Theorem 3**     *1. For every distribution* $P$ *and feature* $f$*, there exists a feature set* $\mathcal{F}$*, such that adding* $f$ *will not impact the fairness of the distribution, e.g.* $U_{adv}(\mathcal{F}) = U_{adv}(\mathcal{F} \cup \{f\})$*.*

2. *There exist distributions* $P$*, features* $f$ *and* $\mathcal{F}'$*, such that* $U_{adv}(\mathcal{F}') = 0$ *and* $U_{adv}(\{f\}) = 0$*, but* $U_{adv}(\mathcal{F}' \cup \{f\}) = 1$ *.*

**Theorem 4**     *1. For any feature* $f$ *and any featureset* $\mathcal{F}$ *we have* $U_{adv}(\mathcal{F}) \leq U_{adv}(\mathcal{F} \cup \{f\})$*. Similarly, if the representation* $\mathcal{F}$ *is* $(\epsilon, \eta)$*-fairness-enabling, the representation* $\mathcal{F} \cup \{f\}$ *is also* $(\epsilon, \eta)$*-fairness-enabling.*

2. *For every distribution* $P$ *and every feature* $f$*, there exists a feature set* $\mathcal{F}$*, such that* $\mathcal{F} \cup \{f\}$ *is* $(\eta, \epsilon)$*-fairness-enabling, if and only if* $\mathcal{F}$ *is* $(\epsilon, \eta)$*-fairness-enabling. Furthermore, there exists a distribution* $P$*, a feature* $f$ *and a feature set* $\mathcal{F}'$*, such that both* $\mathcal{F}'$ *and* $\{f\}$ *are not* $(\epsilon, \eta)$*-fairness-enabling for any* $\epsilon, \eta < \frac{1}{2}$*, but such that* $\mathcal{F}' \cup \{f\}$ *is* (0, 0)*-fairness-enabling.*

While this section focused on fairness with respect to equalized odds, we note that many of these results can be replicated for other notions of fairness. For a more general version of Theorem 3, which takes into account other fairness notions, like demographic parity, we will refer the reader to the Appendix.

# 6 Impossibility of adversarially fair representations with respect to predictive rate parity

We now show that not all acceptable notions of group fairness always allow a adversarially fair representation, even in a single-task setting. One such notion is *predictive rate parity*.

**Definition 6** *(Predictive rate parity (PRP)) A classifier* $h$ *is considered PRP fair w.r.t. to a marginal data distribution* $P$ *and true classification* $t$ *if the random variable* $t(x)$ *is independent of the group membership,* $G(x)$ *given the classification* $h(x)$*. We denote this fairness objective with* $L^{Pred}$*.*

**Theorem 5** *Adversarial fairness w.r.t. $P$ and $L^{Pred}$ is only possible, if $P$ has equal success rates for both groups.*

This theorem results from the fact that the classifier which maps every instance to label $1$ is not fair w.r.t. to $L^{Pred}$ if $P$ does not have equal success rates. The quantitative version of predictive rate parity as well as a more general version of Theorem 5, giving a characterization of adversarial fairness in the case of equal success rates can be found in the appendix.

## 7  Conclusion

In this paper we introduced a general taxonomy of notions of fair representation, taking into consideration both different objectives of decision makers using the representation, and different group fairness notions. Within this taxonomy we showed several impossibility results about fair representation learning.

Our main result addressed the existence of generic fair representations and of fair transfer learning. We show that even seemingly task-independent fairness notions like demographic parity are vulnerable to shifts in marginals between tasks. We conclude the impossibility of having generic data representations that guarantee (even just) DP fairness with respects to tasks whose marginal distributions are not considered when designing the representation. Furthermore, we show that it is impossible to have an adversarially fair representation with respect to several tasks and the equalized odds notion of fairness, if those tasks do not all fulfill statistical parity. These insights stand in contrast to the impression arising from recent papers [10] that claim to learned transferable fair decisions.

We also considered the question of "fairness of a feature", which has been used in legal scenarios. We showed that for notions of decision-making fairness other than demographic parity, the fairness of a single feature is an ill defined notion. Namely, the impact of a feature on the fairness of a decision cannot be determined without considering the other features of the representation.

Lastly, we show that some fairness notions, like predictive rate parity, do not always allow an adversarially fair representation, even if it is just for a single task.

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
