# Appendix

The appendix is organized as follows: We will first give an extended discussion of related work (**A2 Related Work**). We then give a small extension to our taxonomy to deal with sample-based decision makers (**A2 Formal Setup**). We will then give the proofs of Section 4 (**A4 Can there be a generic fair representation?**). We then give an extended version of Section 5, which includes definitions of the assumptions made in Theorem 2 and the proofs of the section (**A5 Fairness of a feature set vs fairness of a feature**). Following this we give the proofs of Section 6 (**A6 Impossibility of adversarially fair representations with respect to predictive rate parity**). Lastly, we give some characterizations in terms of the underlying distribution of the representation fairness notions defined in our setup section(**A7 Characterizations of different notions of fair representation**).

## A2 Related Work

There is an apparent discrepancy between our impossibility results and the long list of papers claiming to achieve fair representations ([16, 11, 15, 7, 4, 12]) . What is the source of that discrepancy? Note that there is a difference in the setup of the problem; in most of the papers that claim positive results about fair representations, the designer of the fair representation has access to the data distribution w.r.t. which the fairness is being evaluated (to the unlabeled marginal distribution for demographic/statistical parity, and to the full labeled distribution for obtaining Odds Equality fairness). However, in this paper, we address the possibility of having a fixed data representation that can be used by learning agents down the road for different classification tasks.

In many cases the assumptions about the connection between the data available at the times of designing the representation and the tasks it will be used for remains implicit, though. For example [16] define their notion of fairness by saying: "We formulate this using the notion of statistical parity, which requires that the probability that a random element from $X^+$ maps to a particular prototype is equal to the probability that a random element from $X^-$ maps to the same prototype" (where $X^+$ and $X^-$ are the two groups w.r.t. which one aims to respect fairness). However, they do not specify what is the meaning of "a random element". The natural interpretation of these terms is that "random" refers to the uniform distribution over the finite set of individuals over which the algorithm selects. In that case, that information varies with each concrete tasks and is not available to the task-independent representation designer. Alternatively, one could interpret those "random" selections as picking uniformly at random from some predetermined large training set (or data repository) that is fixed for all downstream tasks. Such randomness may well be available to the representation designer, but it misses the intention of statistical parity fairness; For example, the fixed training set there are 10,000 individuals from one group and 20,000 from the other group, but when a bank comes to allocate loans they have 30 applicants from the first group and 100 applicants from the other. For the fairness of these loan allocation decisions, the relevant ratio between the groups is 30/100 rather than the 10,000/20,000 ratio available to the representation designer.

Below we refer in some more detail to several well cited papers discussion the design of fair data representations for the use of classification learning by agents using that data.

**Zemel et al 2013 [16]** "our intermediate representation can be used for other classification tasks (i.e., transfer learning is possible)". "We further posit that such an intermediate representation is fundamental to progress in fairness in classification, since it is composable and not ad hoc; once such a representation is established, it can be used in a blackbox fashion to turn any classification algorithm into a fair classifier, by simply applying the classifier to the sanitized representation of the data.". " Hence the mutual information between Z and S is small, and we have accomplished the goal of obfuscating information about the protected group." Here Z is the representation and S is the group membership.

There are two significant issues with this approach. First, as mentioned above, the algorithm generating the representation can have access only to some fixed data distribution. A distribution that is likely to be irrelevant for specific "blackbox fashion" usage by agents addressing specific classification problems down the road. A second issue has to do with the adherence to statistical parity fairness. Any classification based on statistically fairness respecting representation will be inaccurate to the extent that the ground truth and the group membership are correlated.

**Madras et al 2018 [11]** Address the second concern above by saying: "However, assuring that prediction vendors learn only fair predictors complicates the data owner's choice of representation, which must yield predictors that are never unfair but nevertheless have relatively high utility." "We connect common group fairness metrics (demographic parity, equalize odds, and equal opportunity) to adversarial learning by providing appropriate adversarial objective functions for each metric that upper bounds the unfairness of arbitrary downstream classifiers" (center of right columns on page 1). This is exactly what our paper shows to be impossible. How do they overcome this impossibility? In section 4.2 "Learning" - the algorithm requires the labels (the value of Y there) of the training examples. Thus the representation will change from one "downstream" classifier to another and cannot be constructed in an a priory fashion, in contrast to what the paper's introduction seems to imply.

**Song et al 2019 [15]** The focus of this work is algorithmic efficiency of contracting some type of fair representation. The representation building algorithms optimize several objectives that vary with the type of fairness they are set to guarantee and well as with the *'expressiveness'* of the representation. To achieve their fairness goal, the algorithms require access to the target problem data distribution (the marginal for demographic parity and the full labeled distribution for other notions of group fairness). The notion of expressiveness is related to the specific task (or objective) that the algorithms using then representation is aimed to solve. Consequently, the resulting representations are not useful down the road tasks that the representation designer has no samples from.

**Edwards et al 2017 [7]** Propose an algorithm for constructing fair data representation ALFR "The advantage of the latter (ALFR)approach is that the representations can potentially be reused for different tasks, and there is the possibility of a separation of concerns whereby one party is responsible for making the representations fair, and another party is responsible for making the best predictive model....In addition our approach means that the *representations can be used with any classifier*". However, they discuss only Statistical Parity and suffer the same deficiency we described above - the resulting representation fails to guarantee fairness for tasks with data distributing different than the one used to train the representation. Their promise "can potentially be reused for different tasks" can be met only in the very limited case of tasks sharing the training distribution.

**Creager et al 2019 [4]** Focus on Demographic Parity. In the Discussion section they pose two followup problems: "There are two main directions of interest for future work. First is the question of fairness metrics: a wide range of fairness metrics beyond demographic parity have been proposed (Hardt et al., 2016; Pleiss et al., 2017). Understanding how to learn flexibly fair representations with respect to other metrics is an important step in extending our approach. Secondly, robustness to distributional shift presents an important challenge in the context of both disentanglement and fairness" **We prove that both desired extensions are unattainable!**

**Zhao et al [19, 18]** The data representation discussed in these papers are only meant for the given task it is trained over. It is therefore essentially different from the scenario of multi-task fairness providing represnetation that our paper discusses.

**McNamara-Williamson 2019 "costs" [12]** Address a different setup of fair representations. Rather than fixing such a representation for use of down the road classification tasks, they view it as a tool for a data curator to prevent intentional unfairness by an agent using that data *for a specific predetermined task* that the representation building algorithm has access to samples from. Therefore this work is orthogonal to our discussion.

**Oneto et al "transferable" 2019 [13]** The motivation is to learn a representation based on some tasks that will make the fair and accurate learning of new, similar tasks more sample efficient. This has a similar flavor to our notion of accuracy-driven agents. The fairness notion used in this paper is demographic parity though. They consider multi-task in the sens of multiple labelling rules for the same marginal distribution. And they show that the utility transfers while demographic parity is still satisfied. Since the tasks for which such a representations intended all share the same marginal data distribution, the ability to guarantee demographic parity is consistent with our results (although, as mentioned above, such an approach suffers from the restriction of tasks to sharing a fixed marginal and from the potential accuracy costs implied by respecting statistical parity).

## A3 Formal Setup

### Extended taxonomy: $\lambda$-robustness

It is not necessarily sufficient that the requirement of accuracy-driven fairness of a representation is not sufficient to guarantee that accuracy-driven decision makers actually arrive at a fair decision rule, if they do not have access to the underlying distribution $P$ but only to a finite i.i.d sample of the distribution. For this, we would need to further require that any decision with close to optimal accuracy has good fairness. This motivates the definition of $\lambda$-robustness.

**Definition 7** ( $\lambda$-robust fairness). *A feature set $\mathcal{F}$ is considered to be $\lambda$-robustly fair w.r.t. the fairness objective $L^{fair}$ and the distribution $P$, if for every $\alpha \in [0,1]$ every classifier $h \in \mathcal{H}_F$ with $L_P^\alpha(h) \leq \min_{h \in \mathcal{H}_F} L_P^\alpha(h) + \epsilon$ has group unfairness bounded by $L_P^{fair}(h) \leq \min_{h \in \arg\min_{h' \in \mathcal{H}_F} L_P^\alpha(h')} L_P^{fair}(h) + \lambda\epsilon$. Similar to above definition, we say representation $\mathcal{F}$ is $\lambda$-robustly fair w.r.t. to $L^{fair}$, $P$ and $\alpha$ if the guarantee above holds for a particular threshold $\alpha$.*

Note that, if a representation is both accuracy-driven fair and $\lambda$-robust fair, an accuracy-driven decision maker, who bases their decision on an i.i.d. sample of that representation, is guaranteed to arrive at a fair decision rule with high probability over the sample generation, if the sample size is sufficient to guarantee accuracy.

## A4 Can there be a generic fair representation?

### Proofs

**Corollary 1.** *No data representation can guarantee the DP fairness of any non-trivial classifier w.r.t. all possible data generating distributions (over any fixed domain set with any fixed partition into non-empty groups). That is, any non-constant representation F, cannot be adversarially fair with respect to $L^{DP}$ and any arbitrary task $P$.*

**Proof of Corollary 1:** For any non-constant function $f$, we have seen that there exists a marginal $P_X$ such that $f$ does not fulfill demographic parity with respect to $P_X$ (Claim 1). Now if a representation $F$ is non-constant, it allows some non-constant function using that representation. Thus no non-constant representation can fulfill adversarial demographic parity with respect to *any* distribution $P$. $\square$

### Claim: 2.

*For every function non-constant function $f : X \to \{0,1\}$ and every non-constant classifier $h : X \to \{0,1\}$ with $h \neq f$ and $h \neq 1 - f$ (where $1$, denotes the function that maps every element to 1), there exists a marginal $P_X$, such that $h$ has high unfairness with respect to $L^{EO}$ and $P = (P_X, f)$, (i.e. $L_P^{EO}(h) \geq 0.5$).*

*Let $f : X \to \{0,1\}$ be any function and $h : X \to \{0,1\}$ be any classifier such that $h$ is non-committing with respect to ground-truth $f$, i.e. none of the subsets $\{x \in X : f(x) = l_1, h(x) = l_2, G(x) = g\}$ for any $g \in \{A, D\}$ and $l_1, l_2 \in \{0,1\}$ is empty. Then there exists a marginal $P_X$, such that $h$ is arbitrarily unfair with respect to $L^{EO}$ and $P = (P_X, f)$, (i.e. $L_P^{EO}(h) \geq 0.9$).*

**Proof of Claim 2:**

1. Let $f : X \to \{0,1\}$ be any non-constant function and $h : X \to \{0,1\}$ be any non-constant classifier with $h \neq f, 1 - f$. Then we know that at least three of the four sets $\{x \in X : f(x) = 1, h(x) = 0\}$, $\{x \in X : f(x) = 0, h(x) = 1\}$, $\{x \in X : f(x) = 1, h(x) = 1\}$ and $\{x \in X : f(x) = 0, h(x) = 0\}$ are non-empty. Thus two of these three sets, agree on the ground truth. Call them $B_1$ and $B_2$ (and let the remaining set be $B_3$). W.l.o.g. $B_1 = \{s \in X : f(x) = 1, h(x) = 0\}$, $B_2 = \{s \in X : f(x) = 1, h(x) = 1\}$.

   - Case 1: $B_1 \cap A \neq \emptyset$ and $B_2 \cap D \neq \emptyset$. Then we can choose the marginal $P_X$ as $P_X(B_1 \cap A) = 0.5$ and $P_X(B_2 \cap D) = 0.5$. Yielding, $L_P^{EO}(h) = 0.5$
   - Case 2: $B_2 \cap A \neq \emptyset$ and $B_1 \cap D \neq \emptyset$: Analogous to Case 1

- Case 3: there is $G \in \{A, D\}$, such that $B_1 \cap G = B_2 \cap G = \emptyset$. W.l.o.g. $G = A$. Then $B_3 \cap A \neq \emptyset$ and $B_1 \cap D \neq \emptyset$ and $B_2 \cap D \neq \emptyset$. In this case we can choose the marginal as $P_X(A \cap B_3) = 0.5$ and $P_X(D \cap B_1) = 0.5$. Then all elements of $D$ will be misclassified and all elements of $A$ will either be classified correctly or be misclassified in the opposite direction, yielding to high EO unfairness. (In the case where the ground truth labeling is constant on one group, we define the misclassification rate with respect to the label it will not achieve to be zero. Then we get $L_P^{EO}(h) \geq 0.5$.)

2. We can choose the marginal $P_X$ as follows:

$$P_X(\{x \in X : G(x) = A, f(x) = 1, h(x) = 1\}) = 0.25$$

$$P_X(\{x \in X : G(x) = A, f(x) = 1, h(x) = 0\}) = 0$$

$$P_X(\{x \in X : G(x) = A, f(x) = 0, h(x) = 1\}) = 0.25$$

$$P_X(\{x \in X : G(x) = A, f(x) = 0, h(x) = 0\}) = 0$$

$$P_X(\{x \in X : G(x) = D, f(x) = 1, h(x) = 1\}) = 0$$

$$P_X(\{x \in X : G(x) = D, f(x) = 1, h(x) = 0\}) = 0.25$$

$$P_X(\{x \in X : G(x) = D, f(x) = 0, h(x) = 1\}) = 0$$

$$P_X(\{x \in X : G(x) = D, f(x) = 0, h(x) = 0\}) = 0.25$$

The resulting unfairness is $L_P^{EO}(h) = 1$.

$\square$

**Corollary 2.** *No data representation can guarantee EO fairness of any non-constant predictor based on that representation for all "downstream" classification learning tasks. That is, any representation $F$ that is not constant on any group, cannot be adversarially fair with respect to $L^{EO}$ and any arbitrary task $P$. This holds even if one restricts the claim to tasks sharing a fixed marginal data distribution.*

**Proof of Corollary 2:** For any ground truth $f : X \to \{0, 1\}$ and any representation $F : X \to Z$, that allows $h : Z \to \{0, 1\}$ as described in Claim 2, there exists a marginal $P_X$ such that $h$ is highly EO unfair with respect to $(P_X, f)$. Note that as long as $h$ is not constant on either group, we can find $P$, such that the requirements from Claim 2 are fulfilled. Thus the representation is not adversarially fair with respect to $(P_X, f)$ and $L^{EO}$. Thus any sufficiently complex representation cannot guarantee fairness for *every* possible covariate shift. $\square$

**Lemma 1.** *Let $P_1$ and $P_2$ be the distributions defining two different tasks with the same marginal $P_X = P_{1,X} = P_{2,X}$ such that at least one of the tasks does not have equal success rates. Let $h_1, h_2 : X \to \{0, 1\}$ be such that $L_{P_1}(h_1) = L_{P_2}(h_2) = 0$, and assume that tasks are non-negligibly different (namely, $L_{P_1}(h_2) \neq 0$). Then, it cannot be the case that both $h_1$ and $h_2$ are EO fair w.r.t. both $P_1$ and $P_2$.*

**Proof of Lemma 1:**

Let $P_1$ and $P_2$ be the distributions defining two different tasks with the same marginal $P_X = P_{1,X} = P_{2,X}$ such that at least one of the tasks does not have equal success rates. Let $h_1, h_2 : X \to \{0, 1\}$ be such that $L_{P_1}(h_1) = L_{P_2}(h_2) = 0$, and assume that tasks are non-negligibly different (namely, $L_{P_1}(h_2) \neq 0$). The easiest way to verify our claim is to realize that $h_1$ being fair w.r.t. to $P_2$ and equalized odds is equivalent to $h_2$ being fair w.r.t. $P_1$ and predictive rate parity. It is known [10] that this can only be fulfilled if the ground-truth has equal success rates. We will now give a more detailed proof of our claim. For the sake of contradiction let us assume that both $h_1$ and $h_2$ are EO fair w.r.t. both $P_1$ and $P_2$. As $L_{P_1}(h_2) \neq 0$, we know that $\mathbb{P}_{x \sim P_1}[h_2(x) = 1, h_1(x) = 0]$ or $\mathbb{P}_{x \sim P_1}[h_2(x) = 0, h_1(x) = 1]$ to be non-zero. W.l.o.g. assume that $\mathbb{P}_{x \sim P_1}[h_2(x) = 1, h_1(x) = 0] \neq 0$. We then know from $h_2$ fulfilling EO fairness with respect to $P_1$, that

$$\mathbb{P}_{x \sim P_X}[h_2(x) = 1 | G(x) = A, h_1(x) = 0] = \mathbb{P}_{x \sim P_X}[h_2(x) = 1 | G(x) = D, h_1(x) = 0].$$

This implies

$$\frac{\mathbb{P}_{x \sim P_X}[h_1(x) = 1, h_2(x) = 1, G(x) = A]}{\mathbb{P}_{x \sim P_X}[h_1(x) = 0, h_2(x) = 1, G(x) = A]} = \frac{\mathbb{P}_{x \sim P_X}[h_1(x) = 1, h_2(x) = 1, G(x) = D,]}{\mathbb{P}_{x \sim P_X}[h_1(x) = 0, h_2(x) = 1, G(x) = D]}.$$

Thus there exists some $\beta_1$ such that:

$$\mathbb{P}_{x \sim P_X}[h_1(x) = 1, h_2(x) = 1, G(x) = A] = \beta_1 \mathbb{P}_{x \sim P_X}[h_1(x) = 0, h_2(x) = 1, G(x) = A]$$

and

$$\mathbb{P}_{x \sim P_X}[h_1(x) = 1, h_2(x) = 1, G(x) = D,] = \beta_1 \mathbb{P}_{x \sim P_X}[h_1(x) = 0, h_2(x) = 1, G(x) = D].$$

Furthermore from $h_1$ fulfilling EO fairness with respect to $P_2$, we know

$$\mathbb{P}_{x \sim P_X}[h_1(x) = 0 | G(x) = A, h_2(x) = 1] = \mathbb{P}_{x \sim P_X}[h_1(x) = 0 | G(x) = D, h_2(x) = 1].$$

This implies

$$\frac{\mathbb{P}_{x \sim P_X}[h_1(x) = 0, h_2(x) = 0, G(x) = A]}{\mathbb{P}_{x \sim P_X}[h_1(x) = 0, h_2(x) = 1, G(x) = A]} = \frac{\mathbb{P}_{x \sim P_X}[h_1(x) = 0, h_2(x) = 0, G(x) = D,]}{\mathbb{P}_{x \sim P_X}[h_1(x) = 0, h_2(x) = 1, G(x) = D]}.$$

Thus there exists some $\beta_2$ such that:

$$\mathbb{P}_{x \sim P_X}[h_1(x) = 0, h_2(x) = 0, G(x) = A] = \beta_1 \mathbb{P}_{x \sim P_X}[h_1(x) = 0, h_2(x) = 1, G(x) = A]$$

and

$$\mathbb{P}_{x \sim P_X}[h_0(x) = 1, h_0(x) = 1, G(x) = D,] = \beta_2 \mathbb{P}_{x \sim P_X}[h_1(x) = 0, h_2(x) = 1, G(x) = D].$$

Now there are two cases.

- Case 1: $\mathbb{P}_{x \sim P_X}[h_2(x) = 0, h_1(x) = 1] = 0$. Then

  $$\mathbb{P}_{x \sim P_1}[h_1(x) = 0, h_2(x) = 1 | G(x) = A] = \mathbb{P}_{x \sim P_1}[h_1(x) = 0, h_2(x) = 1, G(x) = D] = 0$$

  . We then have

  $$\mathbb{P}_{x \sim P_X}[h_1(x) = 0 | G(x) = A] = \frac{1 + \beta_2}{1 + \beta_1 + \beta_2} = \mathbb{P}_{x \sim P_x}[h_2(x) = 1 | G(x) = D]$$

  . Thus $P_1$ fulfills demographic parity. Similarly,

  $$\mathbb{P}_{x \sim P_X}[h_2(x) = 1 | G(x) = A] = \frac{1 + \beta_1}{1 + \beta_1 + \beta_2} = \mathbb{P}_{x \sim P_x}[h_2(x) = 1 | G(x) = D]$$

  . Which implies that $P_2$ also fulfills demographic parity, contradicting our assumption.

- Case 2: $\mathbb{P}_{x \sim P_X}[h_2(x) = 0, h_1(x) = 1] \neq 0$. Then

  $$\mathbb{P}_{x \sim P_X}[h_1(x) = 1 | G(x) = A, h_2(x) = 0] = \mathbb{P}_{x \sim P_X}[h_1(x) = 1 | G(x) = D, h_2(x) = 0].$$

  – Case 2.1: $\mathbb{P}_{x \sim P_X}[h_2(x) = 0, h_1(x) = 0] = \mathbb{P}_{x \sim P_X}[h_2(x) = 1, h_1(x) = 1] = 0$. In this case $\mathbb{P}_{x \sim P_X}[h_1(x) = 1 | G(x) = A, h_2(x) = 0] = \mathbb{P}_{x \sim P_X}[h_1(x) = 1 | G(x) = D, h_2(x) = 0] = 1$ and $\mathbb{P}_{x \sim P_X}[h_1(x) = 0 | G(x) = A, h_2(x) = 1] = \mathbb{P}_{x \sim P_X}[h_1(x) = 0 | G(x) = D, h_2(x) = 1] = 1$. This implies demographic parity for $P_1$ and for $P_2$ contradicting our initial assumptions.

  – Case 2.2. $\mathbb{P}_{x \sim P_X}[h_2(x) = 0, h_1(x) = 0] \neq 0$ or $\mathbb{P}_{x \sim P_X}[h_2(x) = 1, h_1(x) = 1] \neq 0$. w.l.o.g. $\mathbb{P}_{x \sim P_X}[h_2(x) = 1, h_1(x) = 1] \neq 0$. This implies

  $$\frac{\mathbb{P}_{x \sim P_X}[h_1(x) = 1, h_2(x) = 0, G(x) = A]}{\mathbb{P}_{x \sim P_X}[h_1(x) = 1, h_2(x) = 1, G(x) = A]} = \frac{\mathbb{P}_{x \sim P_X}[h_1(x) = 1, h_2(x) = 0, G(x) = D,]}{\mathbb{P}_{x \sim P_X}[h_1(x) = 1, h_2(x) = 1, G(x) = D]}.$$

  Thus there exists some $\beta_3$ such that:

  $$\mathbb{P}_{x \sim P_X}[h_1(x) = 1, h_2(x) = 0, G(x) = A] = \beta_3 \mathbb{P}_{x \sim P_X}[h_1(x) = 1, h_2(x) = 1, G(x) = A]$$

  and

  $$\mathbb{P}_{x \sim P_X}[h_1(x) = 1, h_2(x) = 0, G(x) = D] = \beta_3 \mathbb{P}_{x \sim P_X}[h_1(x) = 1, h_2(x) = 1, G(x) = D]$$

  . We then have

  $$\mathbb{P}_{x \sim P_X}[h_1(x) = 0 | G(x) = A] = \frac{1 + \beta_2}{1 + \beta_1 + \beta_2 + \beta_1 \beta_3} = \mathbb{P}_{x \sim P_x}[h_2(x) = 1 | G(x) = D]$$

  . Thus $P_1$ fulfills demographic parity. Similarly,

  $$\mathbb{P}_{x \sim P_X}[h_2(x) = 1 | G(x) = A] = \frac{1 + \beta_1}{1 + \beta_1 + \beta_2 + \beta_1 \beta_3} = \mathbb{P}_{x \sim P_x}[h_2(x) = 1 | G(x) = D]$$

  . Which implies that $P_2$ also fulfills demographic parity, contradicting our assumption.

626   □

**Theorem 1.** *There can be no data representation $F$ such that for some $P_1, P_2$ as above, the following criteria simultaneously hold:*

    *1. $F$ is adversarially fair w.r.t. $P_1$ and EO*

    *2. $F$ is adversarially fair w.r.t. $P_2$ and EO*

    *3. $F$ allows for perfect accuracy w.r.t. to $P_1$ and $P_2$, i.e. there are $h_1, h_2$ both expressible over the representation $F$, such that $L_{P_1}(h_1) = L_{P_2}(h_2) = 0$.*

**Proof of Theorem 1:** We note that in order for $F$ being adversarially EO fair with respect to both $P_1$ and $P_2$, both $h_1$ and $h_2$ need to be EO fair with respect to $P_1$ and $P_2$, from Lemma 1, we know that this implies that either $P_1 = P_2$ or that both $P_1$ and $P_2$ have equal success rates. Thus we have proven our claim.   □

## A5 Fairness of a feature set vs. fairness of a feature

**Assumptions for Theorem 2**

We now give an explicit definition for the assumptions made in Theorem 2.

**Non-Triviality properties**

**Definition 8.** *We define the following two non-triviality requirements for a feature:*

    *1. **Non-committing** We will call a feature non-committing if it leaves some ambiguity about label and group membership. That is, a feature $f$ is non-committing if there are two distinct values $y_1$ and $y_2$, such that $f$ assigns each of these values to at least one instance of each $X_{A,0}, X_{A,1}, X_{D,1}, X_{D,0}$. i.e. $f^{-1}(y_1) \cap X_i \neq \emptyset$ and $f^{-1}(y_2) \cap X_i \neq \emptyset$ for every $X_i \in \{X_{A,0}, X_{A,1}, X_{D,1}, X_{D,0}\}$*

    *2. **$k$-anonymity** A feature $f$ is $k$-anonymous if knowing this feature, group-membership and label, will only reveal identity of an individuals up to a set of at least $k$ individuals. Namely, for every combination of value of this feature, group membership and class label, there are either no instances satisfying this combination or there are at least $k$ many such instances.*

Another concept we need in order to prove the theorem are feature-induced cells. A set of features $\mathcal{F} = \{f_1, \ldots, f_n\}$ induces an equivalence relation $\sim_{\mathcal{F}}$, by $x \sim_F$ iff $f_i(x) = f_i(y)$ for all $i = 1, \ldots, n$. We call the equivalence classes with respect to $\sim_{\mathcal{F}}$ cells and denote the set of cells for a featureset $\mathcal{F}$ as $\mathcal{C}_{\mathcal{F}}$.

As this theorem will consider accuracy-driven fairness, we will give a shortly introduce the Bayes optimal classifier. First, we need to consider the scores that are induced by a probability distribution.

We define the *ground truth score function* $s_t : \mathcal{C}_{\mathcal{F}} \to [0, 1]$. $s_t^P(C)$ is the probability, w.r.t. $P$, of $x \in C$ having the true-label 1, i.e., $s_t^P(C) = \mathbb{E}_{x \sim P}[t(x)|x \in C]$. In cases where the distribution is unambiguous we will use the abbreviated notation $s_t$ instead of $s_t^P$.

The predictor in $\mathcal{H}_F$ that minimizes $L_P^\alpha$ is the Bayes Optimal predictor $t_{P,F}^\alpha$ that for a cell $C \in \mathcal{C}_{\mathcal{F}}$ assigns the label 1 if $s_t(C) > \alpha$ and 0 otherwise.

**Additional remarks about Theorem 2**

We have shown in Theorem 2 that for any feature fulfilling some non-triviality requirement, there exists a distribution $P$ and two feature sets such that adding $f$ to the feature sets has opposing effect on the fairness of the representation in terms of accuracy-driven fairness w.r.t. equalized odds. We now want to convince the reader that this can be the case for distributions that are not pathological w.r.t. $f$.

To see this we will now show an example of this happening when both $f$ and $\mathcal{F}$ are adversarially fair w.r.t. $P$: Let the domain $X = \{x_1, x_2, x_3, x_4, x_5, x_6, x_7, x_8, x_9, x_{10}, x_{11}, x_{12}\}$ with $X_{A,1} = \{x_1, x_2, x_3\}, X_{D,1} = \{x_4, x_5, x_6\}, X_{A,0} = \{x_7, x_8, x_9\}$, and $X_{D,0} =$

$\{x_{10}, x_{11}, x_{12}\}$. Furthermore consider the uniform distribution $P$ over $X$, i.e. $P(\{x\}) = \frac{1}{12}$ for every $x \in X$. For the construction of the feature set, we only consider binary features $f_i : X \to \{0, 1\}$. Now let $f$ be defined by $f^{-1}(1) = \{x_1, x_5, x_8, x_{12}\}$. Furthermore, let $\mathcal{F} = \{f_1, f_2, f_3\}$ and $\mathcal{F}' = \{f_1', f_2'\}$ with $f_1^{-1}(1) = \{x_1, x_2, x_3, x_5, x_8, x_{12}\}$, $f_2^{-1}(1) = \{x_1, x_2, x_3, x_5, x_{11}, x_{12}\}$, $f_3^{-1}(1) = \{x_1, x_4 x_5, x_6, x_7, x_{11}\}$, $f_1'^{-1}(1) = \{x_1, x_4, x_7, x_{10}\}$ and $f_2'^{-1}(1) = \{x_1, x_2, x_4, x_5, x_7, x_8, x_{10}, x_{11}\}$. The resulting cells for $\mathcal{F}$ and $\mathcal{F}'$ are $\mathcal{C}_{\mathcal{F}} = \{\{x_1, x_5\}, \{x_2, x_3, x_{12}\}, \{x_8\}, \{x_4, x_6, x_7\}, \{x_9\}, \{x_{10}, x_{11}\}\}$ and $\mathcal{C}_{\mathcal{F}'}\{\{x_1, x_4, x_7, x_{10}\}, \{x_2, x_5, x_8, x_{11}\}, \{x_3, x_6, x_9, x_{12}\}\}$. It is easy to see that $\mathcal{F}'$ and $\{f\}$ are adversarially fair w.r.t. $P$ and $L^{EO}$. Furthermore, we have:

$$U^\alpha_{acc}(\mathcal{F} \cup \{f\}) = \frac{1}{2}|\frac{3}{3} - \frac{2}{3}| + \frac{1}{2}|\frac{2}{3} - \frac{1}{3}| = \frac{1}{3} > 0 = U^\alpha_{acc}(\mathcal{F})$$

and

$$U^\alpha_{acc}(\mathcal{F}' \cup \{f\}) = \frac{1}{2}|\frac{3}{3} - \frac{3}{3}| + \frac{1}{2}|\frac{1}{3} - \frac{1}{3}| = 0 <$$

$$\frac{1}{6} = \frac{1}{2}|\frac{3}{3} - \frac{3}{3}| + \frac{1}{2}|\frac{1}{3} - \frac{0}{3}| = U^\alpha_{acc}(\mathcal{F}').$$

Thus we see that there are indeed features $f$ which are adversarially fair w.r.t. $P$ and equalized odds, for which there is this opposing effect of feature deletion. Moreover, we can give general criteria for $f$ and $P$ which are fulfilled by many pairs $(f, P)$, that are sufficient for showing that the phenomenon from Theorem 2 occurs. In our proof we will show that these critiera are fulfilled when the non-triviality criteria of Definition 7 are fulfilled. We will now state these criteria. In the following let $l \in \{0, 1\}$ denote a label and $G \in \{A, D\}$ a group. The opposing label and group will be denoted by $\bar{l}$ and $\bar{G}$ respectively. A feature $f$ is called "generic" with respect to $P$ if there exist sets $C_1, C_2, C_3 \subset X$ with the following properties.

1. $P(C_1) > P(C_2)$

2. $C_1$ and $C_2$ are separated by the feature $f$, i.e. $C_1 \subset f^{-1}(l')$ and $C_2 \subset f^{-1}(1 - l')$ for some $l' \in \{0, 1\}$

3. $C_1$ and $C_2$ are label-homogeneous for different labels and $C_2$ is group homogeneous, i.e. $C_1 \subset t^{-1}(l)$ and $C_2 \subset X_{G, \bar{l}}$.

4. $C_3$ is not split by the feature, i.e. $C_3 \subset f^{-1}(l'')$ for some $l'' \in \{0, 1\}$

5. $C_3$ has the same majority label as $C_1$, i.e. $P(t^{-1}(l) \cap C_3) \geq P(t^{-1}(\bar{l}) \cap C_3)$

6. The fraction of elements of group $\bar{G}$ and label $\bar{l}$ in $C_3$ is sufficiently big in comparison to $C_2$, i.e. $\frac{P(C_3 \cap X_{G,l})}{P(X_{G,l})} \geq \frac{P(C_2)}{P(X_{G,l})}$.

For pairs $(P, f)$ fulfilling such requirements, one can easily construct two representations $\mathcal{F}$ and $\mathcal{F}'$ such that deleting $f$ in the context of $\mathcal{F}$ will increase unfairness and such that deleting $f$ in the context of $\mathcal{F}'$ decreases unfairness in the spirit of the proof of Theorem 2. We define $\mathcal{F}$ as a representation which separates everything but a cell $C' = C_1 \cup C_2$ by labels. For such a representation $\cup\{f\}$ enables perfect accuracy and therefore perfect fairness. However $\mathcal{F}$ is constructed in a way such that thresholding at 0.5 leads to unfair classification. Furthermore we can define $\mathcal{F}'$ as a representation that separates all but two cells $C' = C_1 \cup C_2$ and $C'' = C_3$ perfectly by labels. By definition, the threshold that cuts at 0.5 on featureset $\mathcal{F}' \cup \{f\}$ is now less fair than the threshold that cuts at 0.5 on featureset $\mathcal{F}'$ as the misclassification rates of $C'$ and $C''$ affect different groups.

**Impact of feature deletion for other group-fairness notions**

We can make an observation similar to Theorem 3 for demographic parity and predictive rate parity (in cases where adversarial fairness w.r.t. $L^{Pred}$ is possible, i.e., when success rates between groups are equal). The proofs use Theorem 11 and Theorem 10 from Section A7.

**Observation 1.** *For every distribution $P$ and feature $f$, there exists a feature set $\mathcal{F}$, such that adding $f$ will not impact the adversarial fairness w.r.t. $L^{DP}$ of the distribution, e.g. $U^{DP}_{adv}(\mathcal{F}) = U^{DP}_{adv}(\mathcal{F} \cap \{f\})$. Furthermore, there exist distributions $P$, features $f$ and $\mathcal{F}'$, such that $U^{DP}_{adv}(\mathcal{F}') = 0$ and $U^{DP}_{adv}(\{f\}) = 0$, but $U^{DP}_{adv}(\mathcal{F}' \cup \{f\}) = 1$.*

**Observation 2.** *For every distribution $P$ and feature $f$, there exists a feature set $\mathcal{F}$, such that adding $f$ will not impact the adversarial fairness w.r.t. $L^{Pred}$ of the distribution, e.g. $U_{adv}^{Pred}(\mathcal{F}) = U_{adv}^{Pred}(\mathcal{F} \cap \{f\})$. Furthermore, there exist distributions $P$, features $f$ and $\mathcal{F}'$, such that $U_{adv}^{Pred}(\mathcal{F}') = 0$ and $U_{adv}^{Pred}(\{f\}) = 0$, but $U_{adv}^{Pred}(\mathcal{F}' \cup \{f\}) = 1$.*

In Section A7, you will also find another observation related to feature deletion, which uses the terminology of that section(Observation ).

## Proofs

**Theorem 2.** *For every 6-anonymous non-committing feature $f$, there exists a probability function $P$ over $X$ and feature sets $\mathcal{F}$ and $\mathcal{F}'$ such that:*

- *The accuracy-driven fairness w.r.t $L^{EO}$, $P$ and $\alpha = 0.5$ of $\mathcal{F} \cup \{f\}$ is greater than that of $\mathcal{F}$, i.e.*

$$U_{acc}^{\alpha}(\mathcal{F} \cup \{f\}) < U_{acc}^{\alpha}(\mathcal{F})$$

  *Thus, deleting $f$ in this context will increase unfairness.*

- *The accuracy-driven fairness w.r.t $L^{EO}$, $P$ and $\alpha = 0.5$ of $\mathcal{F}' \cup \{f\}$ is less than that of $\mathcal{F}'$, i.e.*

$$U_{acc}^{\alpha}(\mathcal{F}' \cup \{f\}) > U_{acc}^{\alpha}(\mathcal{F}')$$

  *Thus, deleting $f$ in this context will decrease unfairness.*

**Proof of Theorem 2:** In the following, we will denote false negative rates and false positive rates of the two groups $A$ and $D$ for the optimal classifier induced by a feature set $\mathcal{F}$ as $FP_{A,\mathcal{F}}$, $FP_{D,\mathcal{F}}$, $FN_{A,\mathcal{F}}$ and $FN_{D,\mathcal{F}}$. The difference of false positive rates with respect to $\mathcal{F}$ will be denoted by $\Delta FP_{\mathcal{F}}$ and the difference in false negative rates with respect to $\mathcal{F}$ will be denoted by $\Delta FN_{\mathcal{F}}$. We will call instances of "A" advantaged and instances of "D" disadvantaged. Furthermore, we call instances with ground-truth label "1" deserving and those with ground truth label "0" undeserving.

For each case will will start by listing sufficient requirements for the effect of deleting $f_i$ being an increase/decrease of features. We find that those scenarios are quite likely to appear in reality. However, when we focus on constructing the examples, we will focus on making the construction as simple as possible, rather than modeling a realistic scenario.

On a high level this proof will construct the two feature sets $\mathcal{F}$ and $\mathcal{F}'$ such that in both cases there exist only three cells that include mislabeled instances. Furthermore the union of those cells for featurization $\mathcal{F}$ and the union of those cells for the featurization $\mathcal{F}'$ will be chosen to be disjoint. This way we can pick $P$ independently for both cases. Furthermore there will be instances of each $X_{A,1}$, $X_{A,0}$, $X_{D,1}$ and $X_{D,0}$ are not in either of those cells. This way we can ensure that most of the probability mass does not sit in either of those cells and therefore control for normalizing factors. This makes it possible to choose $P$ in a way to fulfill all the inequalities.

We first consider the case where deleting $f_i$ increases fairness. We will choose $f_1, \ldots, f_{i-1}$ in such a way that deleting $f_i$ from $\{f_1, \ldots, f_{i-1}, f_i\}$ decreases fairness. We will first give a list of properties that such a set $f_1, \ldots, f_{i-1}$ should fulfill and then argue why they can all be fulfilled simultaneously. We propose $f_1, \ldots, f_{i-1}$ be chosen in a way such that deleting $f_i$ from $f_1, \ldots, f_{i-1}, f_i$ conserves all but two cells $C_i^0$ and $C_i^1$ of the featurization $\{f_1, \ldots, f_{i-1}, f_i\}$. The two cells $C_i^0$ and $C_i^1$ are merged by the deletion of $f_i$ into $C_i$ such that

1. $C_i$ has majority label 1, i.e.

$$P((X_{A,0} \cup X_{D,0}) \cap C_i) < P((X_{A,1} \cup X_{D,1}) \cap C_i)$$

   $C_0^i$ only consists of disadvantaged undeserving and advantaged deserving people

2. $C_i^0$ has majority label 0, i.e.

$$P((X_{A,0} \cup X_{D,0}) \cap C_i^0) > P((X_{A,1} \cup X_{D,1}) \cap C_i^0)$$

3. false positive rate of $A$ is higher than of $D$:

$$A(FP_{\mathcal{F}}) > D(FP_{\mathcal{F}})$$

4. after deleting $f_i$ the false positive rate of $A$ will still be higher than that of $D$:

$$\Delta FP_{\mathcal{F}} > \frac{P(C_0^i \cap X_{D,1})}{P(X_{D,1})} - \frac{P(C_0^i \cap X_{A,1})}{P(X_{A,1})}$$

5. false negative rate of $D$ is higher than of $A$:

$$D(FN_{\mathcal{F}}) > A(FN_{\mathcal{F}})$$

6. after deleting $f_i$ the false negative rate of $D$ will still be higher than that of $A$:

$$\Delta FN_{\mathcal{F}} > \frac{P(C_0^i \cap X_{D,0})}{P(X_{D,0})} - \frac{P(C_0^i \cap X_{A,0})}{P(X_{A,0})}$$

7. switching labels of $C_i^0$ decreases false negative rate of $D$ more than that of $A$:

$$\frac{P(C_i^0 \cap X_{A,1})}{P(X_{A,1})} < \frac{P(C_i^0 \cap X_{D,1})}{P(X_{D,1})}$$

8. switching labels of $C_i^0$ increases false positive rate of $D$ more than that of $A$:

$$\frac{P(C_i^0 \cap X_{A,0})}{P(X_{A,0})} < \frac{P(C_i^0 \cap X_{D,0})}{P(X_{D,0})}$$

We believe these properties to hold for some realistic sets of features $\{f_1, \ldots, f_{i-1}\}$. We will now construct an example for a cell $C_i = C_i^0 \cup C_i^1$, such that these properties are actually fulfilled.

Let $y_1$ and $y_2$ be defined as in the definition of non-committing features. We will define $C_0^i$ as a subset of the pre-image $f_i^{-1}(y_1)$ and $C_1^i$ as a non-empty set that does not contain any elements of the pre-image $f_i^{-1}(y_1)$. This way, we will ensure that deleting the feature $f_i$ will indeed result in the merging of cells $C_0^i$ and $C_1^i$. Let $C_{a,1} = X_{A,1} \cap f^{-1}(y_1)$ and $C_{d,0} = X_{D,0} \cap f^{-1}(y_1)$ and $C_{a,0} = X_{A,0} \cap f^{-1}(y_1)$ and $C_{d,1} = X_{D,1} \cap f^{-1}(y_1)$. According to our non-committing and 6-anonymity assumption on $f_i$ all of these sets are non-empty and contain at least 6 elements. We will only take 3 of those elements for the construction, such that the construction of similar cells for $\mathcal{F}'$ can be done with the other 3 elements and we can assign probability weights to all of those points independently. Let $C_{d,0}^i \subset C_{d,0}$ and $C_{d,1}^i \dot\cup C_{d,1}^{j,0} \dot\cup C_{d,1}^{j,1} \subset C_{d,1}$ and $C_{a,0}^{j,0} \dot\cup C_{a,0}^{j,1} = C_{a,0}^j \subset C_{a,0}$ and choose $P$ such that

- $P(C_{d,0}^i) > P(C_{d,1}^i)$

- $\frac{P(C_{d,0}^i)}{P(X_{D,0})} < \frac{P(C_{a,0}^{j,1})}{P(X_{A,0})}$

- $P(C_{a,0}^{j,1}) < P(C_{d,1}^{j,1})$

- $P(C_{d,1}^{j,0}) < P(C_{a,0}^{j,0})$

One easy way of achieving (2.),(3.) and (4.) is defining $C_0^i = C_{d,1}^i \cup C_{d,0}^i$. We can achieve (1.) by letting $C_1^i$ be defined as a subset of $(X_{A,1} \cap X_{D,1}) \cap (X \setminus f_i^{-1}(y_i))$ such that $P(C_1^i) > P(C_0^i)$. This can be done because of $f_i$ is non-committing. Since one can always introduce more features that separate perfectly by label, we can construct $f_1, \ldots, f_{i-1}$ such that $\{f_1, \ldots, f_{i-1}\}$ induces only three cells with mixed labels: $C_i, C_j^1 := C_{a,0}^{j,1} \cup C_{d,1}^{j,1}$ and $C_j^0 := C_{d,1}^{j,0} \cup C_{a,0}^{j,0}$. Thus the only misclassified instances are in these cells. It is easy to see that this construction achieves (5.)-(8.).

Now we will show that fulfilling (1.)-(8.) implies that deleting $f_i$ makes the optimal classifier based on the featurization more fair with respect to our fairness notion: The only change in false positive and false negative rates are introduced to labeling the elements of $C_i^0$ as 1 instead of 0. Because of (3.) and (4.) the change in false positive rate difference induced by the deletion of $f_i$ is

$$\frac{P(C_0^i \cap X_{A,1})}{P(X_{A,1})} - \frac{P(C_0^i \cap X_{D,1})}{P(X_{D,1})},$$

which is a negative value because of (8.). Because of (5.) and (6.) the change in false negative rate difference induced by the deletion of $f_i$ is

$$\frac{P(C_0^i \cap X_{D,0})}{P(X_{D,0})} - \frac{P(C_0^i \cap X_{A,0})}{P(X_{A,0})},$$

which is a negative value because of (7.). Therefore the deletion of $f_i$ causes both difference in false negative rate and difference in false positive rate to decrease.

We now consider the case where deleting $f_i$ decreases fairness. We will again give a list of criteria, that are sufficient for our claim to hold and then show a more concrete construction that fulfils these criteria. We can construct $f'_1, \ldots, f'_{i-1}$ such that deleting $f_i$ from $f'_1, \ldots, f'_{i-1}, f_i$ only merges two cells $C'^1_i$ and $C'^0_i$ into $C'_i$, such that

1. $C'_i$ has majority label 1, i.e.
$$P((X_{A,0} \cup X_{D,0}) \cap C'_i) < P((X_{A,1} \cup X_{D,1}) \cap C'_i)$$

   $C_0^i$ only consists of disadvantaged undeserving and advantaged deserving people

2. $C'^0_i$ has majority label 0, i.e.
$$P((X_{A,0} \cup X_{D,0}) \cap C'^0_i) > P((X_{A,1} \cup X_{D,1}) \cap C'^0_i)$$

3. false positive rate of $A$ is higher than of $D$:
$$A(FP_{\mathcal{F}'}) > D(FP_{\mathcal{F}'})$$

4. after deleting $f_i$ the false positive rate of $A$ will still be higher than that of $D$:
$$\Delta FP_{F'} > \frac{P(C_0'^i \cap X_{A,1})}{P(X_{A,1})} - \frac{P(C_0'^i \cap X_{D,1})}{P(X_{D,1})}$$

5. false negative rate of $D$ is higher than of $A$: $D(FN_{\mathcal{F}'}) > A(FN_{\mathcal{F}'})$

6. after deleting $f_i$ the false negative rate of $D$ will still be higher than that of $A$:
$$\Delta FN_{\mathcal{F}'} > \frac{P(C_0'^i \cap X_{A,0})}{P(X_{A,0})} - \frac{P(C_0'^i \cap X_{D,0})}{P(X_{D,0})}$$

7. switching labels of $C_i^0$ decreases false negative rate of $A$ more than that of $D$:
$$\frac{P(C_i'^0 \cap X_{D,1})}{P(X_{D,1})} < \frac{P(C_i'^0 \cap X_{A,1})}{P(X_{A,1})}$$

8. switching labels of $C_i'^0$ increases false positive rate of $A$ more than that of $D$:
$$\frac{P(C_i'^0 \cap X_{D,0})}{P(X_{D,0})} < \frac{P(C_i'^0 \cap X_{A,0})}{P(X_{A,0})}$$

We will define $C_0'^i$ as a subset of the preimage $f_i^{-1}(y_2)$ and $C_1'^i$ as a non-empty set that does not contain any elements of the preimage $f_i^{-1}(y_2)$. This way, we will ensure that deleting the feature $f_i$ will indeed result in the merging of cells $C_0'^i$ and $C_1'^i$. Let $C_{d,0}, C_{d,1}, C_{a,0}$ and $C_{a,1}$ be defined as before. Let $C_{a,1}'^i \subset C_{a,1}$ and $C_{a,0}'^i \dot\cup C_{a,0}'^{j,0} \dot\cup C_{a,0}'^{j,1} \subset C_{a,0}$, $C_{d,1}'^{j,0} \dot\cup C_{d,1}'^{j,1} = C_{d,1}^j \subset C_{d,1}$, such that they are disjoint from all the non-perfectly labeled cells in the featurization $\mathcal{F}_1$. Therefore we can choose $P$ in such a way that

- $P(C_{a,0}'^i) > P(C_{a,1}'^i)$

- $\frac{P(C_{a,0}'^i)}{P(X_{A,0})} < \frac{P(C_{a,0}'^{j,1})}{P(X_{A,0})}$

- $P(C_{a,0}'^{j,1}) < P(C_{d,1}'^{j,1})$,

- $P(C_{d,1}'^{j,0}) < P(C_{a,0}'^{j,0})$,

Similar to the above case, a way of achieving (2.),(3.) and (4.) is defining $C_0'^i = C_{a,1}'^i \cup C_{a,0}'^i$. We can achieve (1) by defining $C_i'^1$ as a subset of $(X_{A,1} \cap X_{D,1}) \cap (X \setminus f^{-1}(y_2))$ such that $P(C_i'^1) > P(C_i'^0)$. This can be done because of the second assumption for $f_i$. Since one can always introduce more features that separate perfectly by label, we can construct $f_1, \ldots, f_{i-1}$ such that $\{f_1, \ldots, f_{i-1}\}$ induces only three cells with mixed labels: $C_i'$, $C_j'^1 := C_{a,0}'^{j,1} \cup C_{d,1}'^{j,1}$ and $C_j'^0 := C_{d,1}'^{j,0} \cup C_{a,0}'^{j,0}$. Thus the only misclassified instances are in these cells. It is easy to check that this construction achieves (5.)-(8.).

Because of (3.) and (4.) the change in false positive rate difference induced by the deletion of $f_i$ is

$$\frac{P(C_0'^i \cap X_{A,1})}{P(X_{A,1})} - \frac{P(C_0'^i \cap X_{D,1})}{P(X_{D,1})},$$

which is a positive value because of (8.). Because of (5.) and (6.) the change in false negative rate difference induced by the deletion of $f_i$ is

$$\frac{P(C_0'^i \cap X_{D,0})}{P(X_{D,0})} - \frac{P(C_0'^i \cap X_{A,0})}{P(X_{A,0})},$$

which is a positive value because of (7.). Therefore the deletion of $f_i$ causes both difference in false negative rate and difference in false positive rate to increase.

$\square$

**Theorem 3.**    *1. For every distribution $P$ and feature $f$, there exists a feature set $\mathcal{F}$, such that adding $f$ will not impact the fairness of the distribution, e.g. $U_{adv}(\mathcal{F}) = U_{adv}(\mathcal{F} \cup \{f\})$.*

*2. There exist distributions $P$, features $f$ and $\mathcal{F}'$, such that $U_{adv}(\mathcal{F}') = 0$ and $U_{adv}(\{f\}) = 0$, but $U_{adv}(\mathcal{F}' \cup \{f\}) = 1$.*

**Proof of Theorem 3:**

1. For any distribution $P$ and feature $f$ we can choose a representation $\mathcal{F}$ such that $\mathcal{C}_\mathcal{F} = \mathcal{C}_{\mathcal{F} \cup \{f\}}$. It is obvious that the fairness will not change between those representations.

2. The following example establishes the second claim: Consider the domain $X = x_1, x_2, x_3, x_4, x_5, x_6, x_7, x_8$ with $X_{A,1} = \{x_1, x_2\}$, $X_{D,1} = \{x_3, x_4\}$, $X_{A,0} = \{x_5, x_6\}$ and $X_{D,0} = \{x_7, x_8\}$,. Furthermore let $\mathcal{F} = \{f_1, f_2\}$ with $f_1^{-1}(1) = \{x_1, x_3, x_5, x_7\}$ and $f_2^{-1}(1) = \{x_1, x_4, x_5, x_8\}$. Furthermore let $P$ be uniform over $X$,i.e. $P(\{x_1\}) = P(\{x_2\}) = P(\{x_3\}) = P(\{x_4\}) = P(\{x_5\}) = P(\{x_6\}) = P(\{x_7\}) = P(\{x_8\}) = 0.125$. Thus, we have adversarial fairness w.r.t. EO for both features, i.e.

$$\frac{P(X_{A,1} \cap f_1^{-1}(1))}{P(t^{-1}(1) \cap f_1^{-1}(1))} = \frac{P(\{x_1\})}{P(\{x_1, x_2\})} = 0.5 =$$

$$\frac{P(\{x_3\})}{P(\{x_3, x_4\})} = \frac{P(X_{D,1} \cap f_1^{-1}(1))}{P(t^{-1}(1) \cap f_1^{-1}(1))}.$$

$$\frac{P(X_{A,0} \cap f_1^{-1}(1))}{P(t^{-1}(0) \cap f_1^{-1}(1))} = \frac{P(\{x_5\})}{P(\{x_5, x_6\})} = 0.5 =$$

$$= \frac{P(\{x_7\})}{P(\{x_7, x_8\})} = \frac{P(X_{D,0} \cap f_1^{-1}(1))}{P(t^{-1}(0) \cap f_1^{-1}(1))}.$$

$$\frac{P(X_{A,1} \cap f_2^{-1}(1))}{P(t^{-1}(1) \cap f_2^{-1}(1))} \frac{P(\{x_1\})}{P(\{x_1, x_2\})} = 0.5 =$$

$$= \frac{P(\{x_4\})}{P(\{x_3, x_4\})} = \frac{P(X_{D,1} \cap f_2^{-1}(1))}{P(t^{-1}(1) \cap f_2^{-1}(1))}.$$

$$\frac{P(X_{A,0} \cap f_2^{-1}(1))}{P(t^{-1}(0) \cap f_2^{-1}(1))} \frac{P(\{x_5\})}{P(\{x_5, x_6\})} = 0.5$$

$$= \frac{P(\{x_8\})}{P(\{x_7, x_8\})} = \frac{P(X_{D,0} \cap f_2^{-1}(1))}{P(t^{-1}(0) \cap f_2^{-1}(1))}.$$

However, the featureset $\mathcal{F}$ does not have adversarial fairness w.r.t. EO (using Theorem 6) : $\mathcal{C}_{\mathcal{F}} = \{C_1, C_2, C_3, C_4\}$ with $C_1 = \{x_1, x_5\}$, $C_2 = \{x_2, x_6\}$, $C_3 = \{x_3, x_7\}$, and $C_4 = \{x_4, x_8\}$. Consider the classifier $h \in \mathcal{H}_{\mathcal{F}}$ with $h^{-1}(1) = \{C_1, C_3\}$. Then $L_P^{EO}(h) = \sum_{l \in \{0,1\}} \left| \frac{P(h^{-1}(|1-l|) \cap X_{A,l})}{P(X_{A,l})} - \frac{P(h^{-1}(|1-l|) \cap X_{D,l})}{P(X_{D,l})} \right| = |\frac{1}{2} - 0| + |0 - \frac{1}{2}| = 1$. Thus $U_{adv}^{EO}(\mathcal{F}) = 1$.

$\square$

**Theorem 4.**      *1. For any feature $f$ and any featureset $\mathcal{F}$ we have $U_{adv}(\mathcal{F}) \leq U_{adv}(\mathcal{F} \cup \{f\})$. Similarly, if the representation $\mathcal{F}$ is $(\epsilon, \eta)$-fairness-enabling, the representation $\mathcal{F} \cup \{f\}$ is also $(\epsilon, \eta)$-fairness-enabling.*

    *2. For every distribution $P$ and every feature $f$, there exists a feature set $\mathcal{F}$, such that $\mathcal{F} \cup \{f\}$ is $(\eta, \epsilon)$-fairness-enabling, if and only if $\mathcal{F}$ is $(\epsilon, \eta)$-fairness-enabling. Furthermore, there exists a distribution $P$, a feature $f$ and a feature set $\mathcal{F}'$, such that both $\mathcal{F}'$ and $\{f\}$ are not $(\epsilon, \eta)$-fairness-enabling for any $\epsilon, \eta < \frac{1}{2}$, but such that $\mathcal{F}' \cup \{f\}$ is $(0, 0)$-fairness-enabling.*

**Proof of Theorem 4:**

1. We note that $\mathcal{H}_{\mathcal{F}} \subset \mathcal{H}_{\mathcal{F} \cup \{f\}}$. Thus any $\arg\min_{h \in \mathcal{H}_{\mathcal{F}}} L_P^{EO}(h) \leq \arg\min_{h \in \mathcal{H}_{\mathcal{F} \cup \{f\}}} L_P^{EO}(h)$, proving the inequality for adversarial fairness. Furthermore any $h \in \mathcal{H}_{\mathcal{F}}$ with $\epsilon$ loss and $\eta$ unfairness, is also an element of $h \in \mathcal{H}_{\mathcal{F} \cup \{f\}}$, proving our claim about the fairness-driven case.

2. Similarly to the proof of Theorem 3, given a feature $f$ and a distribution $P$, we can construct a feature set $\mathcal{F}$, such that $\mathcal{C}_{\mathcal{F}} = \mathcal{C}_{\mathcal{F} \cup \{f\}}$. Since this implies that $\mathcal{H}_{\mathcal{F}} = \mathcal{H}_{\mathcal{F} \cup \{\}}$, we get the same fairness-enabling for both distributions.
   Furthermore we can construct the following example to proof the second claim: Consider the domain $X = x_1, x_2, x_3, x_4, x_5, x_6, x_7, x_8$ with $X_{A,1} = \{x_1, x_2\}$, $X_{D,1} = \{x_3, x_4\}$, $X_{A,0} = \{x_5, x_6\}$ and $X_{D,0} = \{x_7, x_8\}$,. Furthermore let $\mathcal{F} = \{f_1, f_2\}$ with $f_1^{-1}(1) = \{x_1, x_3, x_5, x_7\}$ and $f_2^{-1}(1) = \{x_1, x_3, x_6, x_8\}$. Furthermore let $P$ be uniform over $X$, i.e. $P(\{x_1\}) = P(\{x_2\}) = P(\{x_3\}) = P(\{x_4\}) = P(\{x_5\}) = P(\{x_6\}) = P(\{x_7\}) = P(\{x_8\}) = 0.125$. On both $\{f_1\}$ and $\{f_2\}$ there are no classifier $h_1 \in \mathcal{H}_{\{\{\infty\}}$ or $h_2 \in \mathcal{H}_{\{\{\epsilon\}}$ with $L_P^{\alpha}(h_1) < \frac{1}{2}$ or $L_P^{\alpha}(h_2) < \frac{1}{2}$ respectively for any $\alpha \in (0, 1)$. Therefore $\{f_1\}$ and $\{f_2\}$ are both not $(\epsilon, \eta)$-best case fair for any $\epsilon, \eta < \frac{1}{2}$. Furthermore, the classifier $h_3$ defined by $h^{-1}(1) = \{x_1, x_2, x_3, x_4\}$ is element of $\mathcal{H}_{\{f_1\} \cup \{f_2\}}$ and has loss $L_P^{\alpha}(h) = 0$ and unfairness $L_P^{EO}(h) = 0$. Thus $\{f_1\} \cup \{f_2\}$ is $(0, 0)$-fairness-enabling for any $\alpha \in (0, 1)$.

$\square$

**Observation 1.** *For every distribution $P$ and feature $f$, there exists a feature set $\mathcal{F}$, such that adding $f$ will not impact the adversarial fairness w.r.t. $L^{DP}$ of the distribution, e.g. $U_{adv}^{DP}(\mathcal{F}) = U_{adv}^{DP}(\mathcal{F} \cap \{f\})$. Furthermore, there exist distributions $P$, features $f$ and $\mathcal{F}'$, such that $U_{adv}^{DP}(\mathcal{F}') = 0$ and $U_{adv}^{DP}(\{f\}) = 0$, but $U_{adv}^{DP}(\mathcal{F}' \cup \{f\}) = 1$.*

**Proof of Observation 1:** Analogous to the Proof of Observation 3 using Theorem 11 and Observation 3. $\square$

**Observation 2.** *For every distribution $P$ and feature $f$, there exists a feature set $\mathcal{F}$, such that adding $f$ will not impact the adversarial fairness w.r.t. $L^{Pred}$ of the distribution, e.g. $U_{adv}^{Pred}(\mathcal{F}) = U_{adv}^{Pred}(\mathcal{F} \cap \{f\})$. Furthermore, there exist distributions $P$, features $f$ and $\mathcal{F}'$, such that $U_{adv}^{Pred}(\mathcal{F}') = 0$ and $U_{adv}^{Pred}(\{f\}) = 0$, but $U_{adv}^{Pred}(\mathcal{F}' \cup \{f\}) = 1$.*

**Proof of Observation 2:** Analogous to the Proof of Theorem 3 using Theorem 10 and Observation 3. $\square$

## A6 Impossibility of adversarially fair representations with respect to predictive rate parity

**Proofs**

**Theorem 5.** *Adversarial fairness w.r.t. $P$ and $L^{Pred}$ is only possible, if $P$ has equal success rates for both groups.*

**Proof of Theorem 5:** We note that in order to achieve adversarial fairness with respect to any representation, the all-one classifier needs to be fair, as any representation $F$ admits any constant classifier. We furthermore note that the all-one classifier is fair with respect to predictive rate parity if and only if the ground truth has equal success rates. This shows our claim.

□

## A7 Characterizations of different notions of fair representations

In this section we characterize accuracy-driven and adversarial representation fairness w.r.t. the odds equality notion of classification fairness. We will start by introducing a property we call *zero-group knowledge*. It is aimed to prevent an adversary from inferring the group membership from the representation, when given access to the ground-truth labels. To ensure that an adversarial agent won't be able to infer group-membership, one would of course require the representation to have demographic parity. However, in situations where label information is correlated with group membership, demographic parity of all features will hurt classification accuracy. In such cases, zero-group-knowledge might be a better tool for concealing group-information.

We will then see that this property is closely related to adversarial fairness.

**Definition 9** (Zero-group-knowledge). *A representation $F$ has zero-group-knowledge w.r.t. a distribution $P$, if for $x \sim P$, knowing the feature vector $F(x)$ will not reveal more information about the group membership $G(x)$ than knowing just the ground truth, $t(x)$. Namely, $G(x) \perp\!\!\!\perp F(x)|t(x)$.*

It turns out that this property is equivalent to adversarial fairness with respect to equalized odds.

**Theorem 6.** *A representation $F$ has zero-group knowledge w.r.t. $P$ if it has adversarial fairness w.r.t to $P$ and the group-fairness measure $L^{EO}$.*

A similar observation has been made and shown by Zhang et al [17], relating the optimization criteria for the goal of concealing group-membership and preventing unfair classification with respect to equalized odds in a representation learning setting with GANs.

We will now give a characterization of accuracy-driven and worse-case fairness in terms of the conditional distributions given label and group-membership over the cells $\mathcal{C}_{\mathcal{F}}$ of a finite feature set $\mathcal{F}$. In the following we will denote the conditional probabilities given label $l$ and group $G$ as $P_{G,l}$. We will see that a representation is adversarially fair, if and only if the conditional probabilities are aligned. It has already been shown in [18] that if conditional probabilities are aligned over a representation, every classifier based on that representation is fair. We go a step further here, by noting, that this is indeed a necessary condition for adversarial fairness.

**Theorem 7.** *A feature set $\mathcal{F}$ is adversarially fair w.r.t. distribution $P$ if and only if for each cell $C \in \mathcal{C}_{\mathcal{F}}$ and for each $l \in \{0,1\}$ we have $P_{A,l}(C) = P_{D,l}(C)$.*

We now give a similar statement for accuracy enforced fairness. Here, the same statement holds, if instead of considering the probability distributions over the set of cells $\mathcal{C}_{\mathcal{F}}$, we consider the set of cells that results from merging all cells of the same score:

**Definition 10** (Score-induced cells). *For a set of cells $\mathcal{C}_{\mathcal{F}}$, the corresponding set of score-induced cells $\mathcal{C}_{\mathcal{F}_{s_t}}$ is the set of cells that is obtained by merging all cells with the same score together. More formally, each feature set and scoring function, induce an equivalence relation $\sim_{\mathcal{F},s_t}$, such that $x \sim_{\mathcal{F},s_t} y$ if and only if there are cells $C_x, C_y \in \mathcal{C}_{\mathcal{F}}$ such that $x \in C_x, y \in C_y$ and $s_t(C_x) = s_t(C_y)$. The set $\mathcal{C}_{\mathcal{F}_{s_t}}$ is then defined as the set of $\sim_{\mathcal{F},s_t}$ equivalence classes.*

**Theorem 8.** *A feature set $\mathcal{F}$ is accuracy-driven fair w.r.t. distribution $P$ if and only if for each cell in the score-induced $C \in \mathcal{C}_{\mathcal{F}_{s_t}}$ and for each $l \in \{0,1\}$ we have $P_{A,l}(C) = P_{D,l}(C)$.*

We can now bound the unfairness in terms of accuracy-driven and adversarial fairness of a representation by the distribution distance of conditional probabilities. For this we take the $\mathcal{H}$-distance as introduced by [2].

**Definition 11** ($\mathcal{H}$-distance). *Given two distributions $P$ and $Q$ over $X$, we define their $\mathcal{H}$-distance by*

$$d_{\mathcal{H}}(P, Q) = \sup_{1_B \in \mathcal{H}} |P(B) - Q(B)|,$$

*where $1_B$ denotes the indicator function of set $B$.*

In the following let $\mathcal{H}_{\mathcal{C}_{\mathcal{F}}, s_t}^{thres} = \{h : \mathcal{C}_{\mathcal{F}} \to \{0, 1\} : \text{for some } \alpha, \ h(C) = 0 \ iff \ s_t(C) < \alpha\}$ be the class of all classifiers that are a threshold in the ground-truth scoring. We can now state a quantitative theorem about the relation between the conditional alignment and the fairness of a representation:

**Theorem 9.** *We can bound adversarial fairness and accuracy enforced fairness of a feature set $\mathcal{F}$ w.r.t. $P$ and $L^{EO}$ by the $d_{\mathcal{H}_{\mathcal{F}}}$-difference and $d_{\mathcal{H}_{\mathcal{C}_{\mathcal{F}}, s_t}^{thres}}$-difference of conditional distributions respectively:*

$$U_{adv}(\mathcal{F}) \leq \frac{1}{2} d_{\mathcal{H}_{\mathcal{F}}}(P_{A,1}, P_{D,1}) + \frac{1}{2} d_{\mathcal{H}_{\mathcal{F}}}(P_{A,0}, P_{D,0})$$

$$U_{acc}(\mathcal{F}) \leq \frac{1}{2} d_{\mathcal{H}_{\mathcal{C}_{\mathcal{F}}, s_t}^{thres}}(P_{A,1}, P_{D,1}) + \frac{1}{2} d_{\mathcal{H}_{\mathcal{C}_{\mathcal{F}}, s_t}^{thres}}(P_{A,0}, P_{D,0})$$

*Furthermore, we can lower bound the adversarial fairness of a representation by*

$$\frac{1}{2} d_{\mathcal{H}_{\mathcal{F}}}(P_{A,l}, P_{D,l}) \leq U_{adv}(\mathcal{F})$$

*for every $l \in \{0, 1\}$*

Note that for both bounds there exist probability distributions $P$ such that equality holds in all cases. Furthermore we note that since the $\mathcal{H}$-distance between two distributions can be estimated, if $\mathcal{H}$ has a finite VC-dimension [2], we can estimate both the upper and the lower bound with a sample size dependent on $|\mathcal{C}_{\mathcal{F}}|$, when given access to i.i.d. samples from $P_{A,1}, P_{D,1}, P_{D,0}$ and $P_{A,0}$ each.

From Theorem 5 we know that there are distributions for which there is no representation that has adversarial fairness with respect to predictive rate parity. In cases, where such a adversarial representation is achievable, however, we can characterize it by the following natural requirement on the representation, as we will see in the following theorem.

**Definition 12.** *A feature set $\mathcal{F}$ has calibration parity w.r.t. a distribution $P$ if for every cell $C \in \mathcal{C}_{\mathcal{F}}$ both groups have equal success probability. Equivalently, one can say that for a random instance $x \in P$ the ground truth labeling $t(x)$ and the group membership $G(x)$ are statistically independent, when the feature vector $F(x)$ of $x$ is known, i.e. $G(x) \perp\!\!\!\perp t(x)|F(x)$.*

**Theorem 10.** *A feature set $\mathcal{F}$ has calibration parity w.r.t. $P$ if it has adversarial fairness w.r.t $P$ and the group-fairness measure $L^{Pred}$. The other direction does not hold. In particular, adversarial fairness w.r.t. $P$ and $L^{Pred}$ is only possible, if $P$ has equal success rates for both groups*

**Theorem 11.** *A feature set $\mathcal{F}$ has demographic parity w.r.t. $P$ if and only if it has adversarially fair w.r.t $P$ and the group-fairness objective $L^{DP}$.*

### Impact of a feature on fairness for other group fairness notions

We can make another observation about the impact of feature deletion on unfairness for other notions of group fairness.

**Observation 3.**
- *There exists a distribution $P$ and a feature set $\mathcal{F}$ such that each $f \in \mathcal{F}$ the feature set $\{f\}$ has zero-group-knowledge w.r.t. $P$, but $\mathcal{F}$ is not and $U_{adv}(\mathcal{F}) = 1$*

- *There exists a distribution $P$ and a feature set $\mathcal{F}$ such that each $f \in \mathcal{F}$, the feature set $\{f\}$ has demographic parity w.r.t. $P$, but $\mathcal{F}$ has not. Furthermore the group-membership can be perfectly determined by $\mathcal{F}$, i.e. for every cell $C \in \mathcal{C}_{\mathcal{F}}$ we have*

$$\mathbb{P}_{x \sim P}[x \in A | x \in C] \in \{0, 1\}$$

- *There exists a distribution $P$ and a feature set $\mathcal{F}$ such that each $f \in \mathcal{F}$, the feature set $\{f\}$ has calibration parity w.r.t. $P$, but $\mathcal{F}$ has not. Furthermore the scores for the different groups in each cell are perfectly opposed, i.e. $C \subseteq A$ or $C \subseteq D$.*

## Proofs

**Theorem 6.** *A feature set $\mathcal{F}$ has zero-group knowledge w.r.t. $P$ if it has adversarial fairness w.r.t to $P$ and the group-fairness measure $L^{EO}$.*

**Proof of Theorem 6:**

$$\frac{P(h^{-1}(1) \cap X_{G,l})}{P(X_{G,l})} = \frac{\sum_{C \in \mathcal{C}_{\mathcal{F}}:C \in h^{-1}(1)} P(h^{-1}(1) \cap X_{G,l})}{P(X_{G,l})} = \sum_{C \in \mathcal{C}_{\mathcal{F}}:C \in h^{-1}(1)} \frac{P(C \cap X_{G,l})}{P(X_{G,l})}$$

$$= \sum_{C \in \mathcal{C}_{\mathcal{F}}:P(C \in h^{-1}(1)} \frac{P(C \cap t^{-1}(l))}{P(t^{-1}(l))} = \frac{P((h^{-1}(1) \cap t^{-1}(l))}{P(t^{-1}(l))}$$

Thus any hypothesis $h \in \mathcal{H}_{\mathcal{F}}$ is fair w.r.t. to the odds equality notion of fairness.

Assume $\mathcal{F}$ does not have zero-group-knowledge. Thus $F(x)$ and $G(x)$ are dependent given the ground truth $t(x)$. Thus there exists label $l \in \{0,1\}$, group $G \in \{A, D\}$ and a cell $C \in \mathcal{C}_{\mathcal{F}}$ with $\frac{P(C \cap X_{G,l})}{P(X_{G,l})} \neq \frac{P(C \cap t^{-1}(l))}{P(t^{-1}(l))}$. Now consider the hypothesis class $h$ defined by $h^{-1}(1) = C$. For this hypothesis we have $\frac{P(h^{-1}(1) \cap X_{G,l})}{P(X_{G,l})} \neq \frac{P(h^{-1}(1) \cap t^{-1}(l))}{P(t^{-1}(l))}$. Thus, not every hypothesis $h \in \mathcal{H}_{\mathcal{F}}$ fulfills equalized odds.

$\square$

**Theorem 7.** *A feature set $\mathcal{F}$ is adversarially fair w.r.t. distribution $P$ if and only if for each cell $C \in \mathcal{C}_{\mathcal{F}}$ and for each $l \in \{0,1\}$ we have $P_{A,l}(C) = P_{D,l}(C)$.*

**Proof of Theorem 7:**

Assume $\mathcal{F}$ is adversarially fair w.r.t. to $P$ and $L^{EO}$. This means that every $h \in \mathcal{H}_{\mathcal{F}}$ is fair w.r.t to $L^{EO}$. Now take any cell $C \in \mathcal{C}_{\mathcal{F}}$ and let $h$ be defined by $h^{-1}(1) = C$. Then we know that $\frac{P(X_{A,1} \cap C)}{P(X_{A,1})} = \frac{P(X_{D,1} \cap C)}{P(X_{D,1})}$ and $\frac{P(X_{A,0} \cap C)}{P(X_{A,0})} = \frac{P(X_{D,0} \cap C)}{P(X_{D,0})}$. Thus, for each $l \in \{0,1\}$ we have $P_{A,l}(C) = P_{D,l}(C)$.

Now assume, we have for each $l \in \{0,1\}$ we have $P_{A,l}(C) = P_{D,l}(C)$. Then for any $h \in \mathcal{H}_{\mathcal{F}}$, we get

$$\frac{P(X_{A,l} \cap h^{-1}(1))}{P(X_{A,l})} = \sum_{C \in h^{-1}(1)} \frac{P(X_{A,l} \cap C)}{P(X_{A,l})} =$$

$$= \sum_{C \in h^{-1}(1)} \frac{P(X_{D,l} \cap C)}{P(X_{D,l})} = \frac{P(X_{D,l} \cap h^{-1}(1))}{P(X_{D,l})}.$$

Thus $L_P^{EO}(h) = 0$. $\square$

**Theorem 8.** *A feature set $\mathcal{F}$ is accuracy-driven fair w.r.t. distribution $P$ if and only if for each cell in the score-induced $C \in \mathcal{C}_{\mathcal{F}_{s_t}}$ and for each $l \in \{0,1\}$ we have $P_{A,l}(C) = P_{D,l}(C)$.*

**Proof of Theorem 8:** "Conditional probabilities over score-cells align" implies "representation is accuracy-driven fair": We know for every cell $C \in \mathcal{C}_{\mathcal{F}_{s_t}}$

$$\mathbb{P}_{x \sim P}[x \in C | x \in A, t(x) = 0] = \mathbb{P}_{x \sim P}[x \in C | x \in D, t(x) = 0]$$

and

$$\mathbb{P}_{x \sim P}[x \in C | x \in A, t(x) = 1] = \mathbb{P}_{x \sim P}[x \in C | x \in D, t(x) = 1].$$

Thus for every threshold $\alpha \in [0,1]$, we have $\mathbb{P}_{x \sim P}[t_{P,F}^{\alpha}(x) | x \in A, t(x) = 1] = \mathbb{P}_{x \sim P}[t_{P,F}^{\alpha}(x) | x \in D, t(x) = 1]$ and $\mathbb{P}_{x \sim P}[t_{P,F}^{\alpha}(x) | x \in A, t(x) = 0] = \mathbb{P}_{x \sim P}[x \in C | x \in D, t(x) = 0]$. This implies equal false-positive and false-negative rates and therefore group fairness.

" conditional probabilities over score-cells do not align" implies "representation is not accuracy-driven fair": We assume that the conditional probabilities over score induced cells are not aligned.

Let $C_{\mathcal{F}score} = \{C_1, \ldots C_{k'}\}$ such that $s_t(C_i) < s_t(C_j)$ for every $i < j$. Thus, $C_i \in C_{\mathcal{F}s_t}$ with $\mathbb{P}_{x \sim P}[x \in C| \in A, t(x) = 0] \neq \mathbb{P}_{x \sim P}[x \in C_i | x \in D, t(x) = 0]$ or $\mathbb{P}_{x \sim P}[x \in C_i | \in A, t(x) = 1] \neq \mathbb{P}_{x \sim P}[x \in C_i | x \in D, t(x) = 1]$. Now consider the threshold classifier with threshold $s_t(C_i)$. We can consider two cases:

- Case 1: $\dfrac{P(t_{P,F}^{s(C_i)-1}(0) \cap X_{A,1})}{P(X_{A,1})} \neq \dfrac{P(t_{P,F}^{s(C_i)-1}(0) \cap X_{D,1})}{P(X_{D,1})}$ or $\dfrac{P(t_{P,F}^{s(C_i)-1}(1) \cap X_{A,0})}{P(X_{A,0})} \neq$

  $\dfrac{P(t_{P,F}^{s(C_i)-1}(1) \cap X_{D,0})}{P(X_{D,0})}$. This implies $L_{F,P}^{fair}(t_{P,F}^{s(C_i)}) > 0$. In this case, the Bayes classifier that cuts at $s_t(C_i)$ is unfair. Thus there exist a threshold classifier that is unfair.

- Case 2: $\dfrac{P(t_{P,F}^{s(C_i)-1}(0) \cap X_{A,1})}{P(X_{A,1})} = \dfrac{P(t_{P,F}^{s(C_i)-1}(0) \cap X_{D,1})}{P(X_{D,1})}$ and $\dfrac{P(t_{P,F}^{s(C_i)-1}(1) \cap X_{A,0})}{P(X_{A,0})} =$

  $\dfrac{P(t_{P,F}^{s(C_i)-1}(1) \cap X_{D,0})}{P(X_{D,0})}$.

  However since $\mathbb{P}_{x \sim P}[x \in C_i | \in A, t(x) = 0] \neq \mathbb{P}_{x \sim P}[x \in C_i | x \in D, t(x) = 0]$ or $\mathbb{P}_{x \sim P}[x \in C_i | \in A, t(x) = 1] \neq \mathbb{P}_{x \sim P}[x \in C_i | x \in D, t(x) = 1]$, this implies that $i > 1$. Now consider the threshold classifier with threshold $s_t(C_{i-1})$: $\mathbb{P}_{x \sim P}[t^{s(C_{i-1})}(x) = 0| \in A, t(x) = 1]$
  $= \mathbb{P}_{x \sim P}[t^{s(C_i)}(x) = 0| \in A, t(x) = 1]$
  $+ \mathbb{P}_{x \sim P}[x \in C| \in A, t(x) = 1]$
  $\neq \mathbb{P}_{x \sim P}[t^{s(C_i)}(x) = 0| \in D, t(x) = 1]$
  $+ \mathbb{P}_{x \sim P}[x \in C| \in D, t(x) = 1]$
  $= \mathbb{P}_{x \sim P}[t^{s(C_{i-1})}(x) = 0| \in D, t(x) = 1]$ or $\mathbb{P}_{x \sim P}[t^{s(C_{i-1})}(x) = 1| \in A, t(x) = 0]$
  $= \mathbb{P}_{x \sim P}[t^{s(C_i)}(x) = 1| \in A, t(x) = 0]$
  $- \mathbb{P}_{x \sim P}[x \in C| \in A, t(x) = 0]$
  $\neq \mathbb{P}_{x \sim P}[t^{s(C_i)}(x) = 0| \in D, t(x) = 0]$
  $- \mathbb{P}_{x \sim P}[x \in C| \in D, t(x) = 1]$
  $= \mathbb{P}_{x \sim P}[t^{s(C_{i-1})}(x) = 1| \in D, t(x) = 0]$ Which implies $L_P^{EO}(t^{s(C_{i-1})}) > 0$. Thus there exist a threshold classifier that is unfair.

$\square$

**Theorem 9.** *We can bound adversarial fairness and accuracy enforced fairness of a feature set $\mathcal{F}$ w.r.t. $P$ and $L^{EO}$ by the $d_{C_{\mathcal{F}}}$-difference and $d_{C_{\mathcal{F}s_t}}$-difference of conditional distributions respectively:*

$$U_{adv}(\mathcal{F}) \leq \frac{1}{2} d_{\mathcal{H}_{\mathcal{F}}}(P_{A,1}, P_{D,1}) + \frac{1}{2} d_{\mathcal{H}_{\mathcal{F}}}(P_{A,0}, P_{D,0})$$

$$U_{acc}(\mathcal{F}) \leq \frac{1}{2} d_{\mathcal{H}_{C_{\mathcal{F},s_t}}^{thres}}(P_{A,1}, P_{D,1}) + \frac{1}{2} d_{\mathcal{H}_{C_{\mathcal{F},s_t}}^{thres}}(P_{A,0}, P_{D,0})$$

*Furthermore, we can lower bound the adversarial fairness of a representation by*

$$\frac{1}{2} d_{\mathcal{H}_F}(P_{A,l}, P_{D,l}) \leq U_{adv}(\mathcal{F})$$

*for every $l \in \{0, 1\}$*

**Proof of Theorem 9:** $U_{adv}(\mathcal{F}) = \max_{h \in \mathcal{H}_{C_{\mathcal{F}}}} L_P^{EO}(h)$
$= \max_{h \in \mathcal{H}_F} \sum_{l \in \{0,1\}} \frac{1}{2} \left| \frac{P(h^{-1}(1-l) \cap X_{A,l})}{P(X_{A,l})} - \frac{P(h^{-1}(1-l) \cap X_{D,l})}{P(X_{D,l})} \right|$
$\leq \frac{1}{2} \sup_{1_B \in \mathcal{H}_F} |P_{A,1}(B) - P_{D,1}(B)| + \frac{1}{2} \sup_{1_B \in \mathcal{H}_F} |P_{A,0}(B) - P_{D,1}(B)|$
$= \frac{1}{2} d_{\mathcal{H}_F}(P_{A,1}, P_{D,1}) + \frac{1}{2} d_{\mathcal{H}_F}(P_{A,0}, P_{D,0})$

$U_{acc}(\mathcal{F}) \leq \max_{h \in \mathcal{H}_F} L_P^{EO}(h)$
$= \max_{h \in \mathcal{H}_F} \sum_{l \in \{0,1\}} \frac{1}{2} \left| \frac{P(h^{-1}(1-l) \cap X_{A,l})}{P(X_{A,l})} - \frac{P(h^{-1}(1-l) \cap X_{D,l})}{P(X_{D,l})} \right|$
$\leq \frac{1}{2} \sup_{1_B \in \mathcal{H}_{C_{\mathcal{F},s_t}}^{thres}} |P_{A,1}(B) - P_{D,1}(B)|$
$+ \frac{1}{2} \sup_{1_B \in \mathcal{H}_{C_{\mathcal{F},s_t}}^{thres}} |P_{A,0}(B) - P_{D,1}(B)|$
$= \frac{1}{2} d_{\mathcal{H}_{C_{\mathcal{F},s_t}}^{thres}}(P_{A,1}, P_{D,1}) + \frac{1}{2} d_{\mathcal{H}_{C_{\mathcal{F},s_t}}^{thres}}(P_{A,0}, P_{D,0})$ Furthermore, for any label $l' \in \{0, 1\}$, we get

$U_{adv}(\mathcal{F}) = \max_{h \in \mathcal{H}_F} L_P^{EO}(h)$
$= \max_{h \in \mathcal{H}_F} \sum_{l \in \{0,1\}} \frac{1}{2} \left| \frac{P(h^{-1}(1-l) \cap X_{A,l})}{P(X_{A,l})} - \frac{P(h^{-1}(1-l) \cap X_{D,l})}{P(X_{D,l})} \right|$

1028 $\geq \frac{1}{2}\max_{h\in\mathcal{H}_F}|\frac{P(h^{-1}(1-l')\cap X_{A,l})}{P(X_{A,l'})} - \frac{P(h^{-1}(1-l')\cap X_{D,l})}{P(X_{D,l'})}|$

1029 $= \frac{1}{2}\sup_{1_B\in\mathcal{H}_F}|P_{A,l'}(B) - P_{D,l'}(B)|$

1030 $= \frac{1}{2}d_{\mathcal{H}_F}(P_{A,l}, P_{D,l})$

1031 $\square$

1032 **Theorem 10.** *A feature set $\mathcal{F}$ has calibration parity w.r.t. $P$ if it has adversarial fairness w.r.t $P$*
1033 *and the group-fairness measure $L^{Pred}$. The other direction does not hold. In particular, adversarial*
1034 *fairness w.r.t. $P$ and $L^{Pred}$ is only possible, if $P$ has equal success rates for both groups*

1035 **Proof of Theorem 10:** Assume $\mathcal{F}$ does not have calibration parity. Thus $t(x)$ and $G(x)$ are
1036 dependent given a feature vector. Thus there exists label $l \in \{0,1\}$, group $G \in \{A, D\}$ and a
1037 cell $C \in \mathcal{C}_\mathcal{F}$ with $\frac{P(C\cap X_{G,l})}{P(C\cap G)} \neq \frac{P(C\cap t^{-1}(l))}{P(C)}$. Now consider the hypothesis class $h$ defined by
1038 $h^{-1}(1) = C$. For this hypothesis we have $\frac{P(h^{-1}(1)\cap X_{G,l})}{P(h^{-1}(l)\cap G)} \neq \frac{P(h^{-1}(1)\cap t^{-1}(l))}{P(h^{-1}(l))}$. Thus, not every
1039 hypothesis $h \in \mathcal{H}_\mathcal{F}$ fulfills predictive rate parity.

1040 The reverse statement is not true. Let $\mathcal{C}_\mathcal{F} = \{C_1, C_2\}$ be such that $P(C_1\cap X_{A,1}) = 0.5, P(C_1\cap$
1041 $X_{A,0}) = 0.1, P(C_1\cap X_{D,1}) = 0.2, P(C_1\cap X_{A,1}) = 0.04$ and $P(C_2\cap X_{A,1}) = P(C_2\cap X_{A,0}) =$
1042 $P(C_1\cap X_{D,1}) = P(C_1\cap X_{A,1}) = 0.04$. The classifier $h$ defined by $h(C) = 1$ for every $C \in \mathcal{C}_\mathcal{F}$
1043 does not have predictive rate parity, since $\frac{P(h^{-1}(1)\cap X_{A,0})}{P(h^{-1}\cap A)} = \frac{P(X_{A,0})}{P(A)} = \frac{14}{68} \neq \frac{22}{100} = P(t^{-1}(1)) =$
1044 $\frac{P(h^{-1}(1)\cap X_{A,0})}{P(h^{-1}\cap A)} = \frac{P(X_{A,0})}{P(A)}$. Moreover, adversarial predictive parity is only possible in cases where
1045 success rates are equal, since unequal success rates always implies that the classifier $h$ defined by
1046 $h(C) = 1$ for every $C \in \mathcal{C}_\mathcal{F}$ does not fulfill predictive rate parity. $\square$

1047 **Theorem 11.** *A feature set $\mathcal{F}$ has demographic parity w.r.t. $P$ if and only if it has adversarially fair*
1048 *w.r.t $P$ and the group-fairness objective $L^{DP}$.*

1049 **Proof of Theorem 11:**

- **Demographic Parity:** Assume $\mathcal{F}$ has demographic parity, then we have for every cell $C \in \mathcal{C}_\mathcal{F}$: $\frac{P(A\cap C)}{P(C)} = P(A)$. Thus, we have for any $h \in \mathcal{H}_\mathcal{F}$:

$$\frac{P(h^{-1}(1)\cap A)}{P(h^{-1}(1))} = \frac{\sum_{C\in\mathcal{C}_\mathcal{F}:C\in h^{-1}(1)} P(C\cap A)}{P(h^{-1}(1))}$$

$$= \frac{\sum_{C\in\mathcal{C}_\mathcal{F}:C\in h^{-1}(1)} P(C)P(A)}{P(h^{-1}(1))} = \frac{P(A)\sum_{C\in\mathcal{C}_\mathcal{F}:C\in h^{-1}(1)} P(C)}{P(h^{-1}(1))} = \frac{P(A)P(h^{-1}(1))}{P(h^{-1}(1))} = P(A)$$

1050 Thus any $h \in \mathcal{H}_\mathcal{F}$ also has demographic parity.
1051

1052 - Assume $\mathcal{F}$ does not have demographic parity. Thus, there exists at least one cell $C \in \mathcal{C}_\mathcal{F}$
1053 with $\frac{P(A\cap C)}{P(C)} \neq P(C)$. Now consider the hypothesis class $h$ defined by $h^{-1}(1) = C$. For
1054 this hypothesis we have $\frac{P(h^{-1}(1)\cap A)}{P(h^{-1}(1))} \neq P(A)$. Thus, not every hypothesis $h \in \mathcal{H}_\mathcal{F}$ has
1055 demographic parity.

1056 $\square$

1057 **Observation 3.** - *There exists a distribution $P$ and a feature set $\mathcal{F}$ such that each $f \in \mathcal{F}$ the*
1058 *feature set $\{f\}$ has zero-group-knowledge w.r.t. $P$, but $\mathcal{F}$ is not and $U_{adv}(\mathcal{F}) = 1$*

- *There exists a distribution $P$ and a feature set $\mathcal{F}$ such that each $f \in \mathcal{F}$, the feature set $\{f\}$*
  *has demographic parity w.r.t. $P$, but $\mathcal{F}$ has not. Furthermore the group-membership can be*
  *perfectly determined by $\mathcal{F}$, i.e. for every cell $C \in \mathcal{C}_\mathcal{F}$ we have*

$$\mathbb{P}_{x\sim P}[x \in A|x \in C] \in \{0,1\}$$

1059 - *There exists a distribution $P$ and a feature set $\mathcal{F}$ such that each $f \in \mathcal{F}$, the feature set*
1060 *$\{f\}$ has calibration parity w.r.t. $P$, but $\mathcal{F}$ has not. Furthermore the scores for the different*
1061 *groups in each cell are perfectly opposed, i.e. $C \subseteq A$ or $C \subseteq D$.*

**Proof of Observation 3:**

- **(zero-group-knowledge)** Consider the domain $X = x_1, x_2, x_3, x_4, x_5, x_6, x_7, x_8$ with $X_{A,1} = x_1, x_2$, $X_{D,1} = x_3, x_4$, $X_{A,0} = x_5, x_6$ and $X_{D,0} = x_7, x_8$,. Furthermore let $\mathcal{F} = \{f_1, f_2\}$ with $f_1^{-1}(1) = \{x_1, x_3, x_5, x_7\}$ and $f_2^{-1}(1) = \{x_1, x_4, x_5, x_8\}$. Furthermore let $P$ be uniform over $X$, i.e. $P(\{x_1\}) = P(\{x_2\}) = P(\{x_3\}) = P(\{x_4\}) = P(\{x_5\}) = P(\{x_6\}) = P(\{x_7\}) = P(\{x_8\}) = 0.125$. Thus, we have zero-group-knowledge for both features, i.e.

$$\frac{P(X_{A,1} \cap f_1^{-1}(1))}{P(t^{-1}(1) \cap f_1^{-1}(1))} = \frac{P(\{x_1\})}{P(\{x_1, x_2\})} = 0.5$$

$$= \frac{P(\{x_3\})}{P(\{x_3, x_4\})} = \frac{P(X_{D,1} \cap f_1^{-1}(1))}{P(t^{-1}(1) \cap f_1^{-1}(1))}.$$

$$\frac{P(X_{A,0} \cap f_1^{-1}(1))}{P(t^{-1}(0) \cap f_1^{-1}(1))} = \frac{P(\{x_5\})}{P(\{x_5, x_6\})} = 0.5$$

$$= \frac{P(\{x_7\})}{P(\{x_7, x_8\})} = \frac{P(X_{D,0} \cap f_1^{-1}(1))}{P(t^{-1}(0) \cap f_1^{-1}(1))}.$$

$$\frac{P(X_{A,1} \cap f_2^{-1}(1))}{P(t^{-1}(1) \cap f_2^{-1}(1))} \frac{P(\{x_1\})}{P(\{x_1, x_2\})} = 0.5$$

$$= \frac{P(\{x_4\})}{P(\{x_3, x_4\})} = \frac{P(X_{D,1} \cap f_2^{-1}(1))}{P(t^{-1}(1) \cap f_2^{-1}(1))}.$$

$$\frac{P(X_{A,0} \cap f_2^{-1}(1))}{P(t^{-1}(0) \cap f_2^{-1}(1))} \frac{P(\{x_5\})}{P(\{x_5, x_6\})} = 0.5$$

$$= \frac{P(\{x_8\})}{P(\{x_7, x_8\})} = \frac{P(X_{D,0} \cap f_2^{-1}(1))}{P(t^{-1}(0) \cap f_2^{-1}(1))}.$$

However, the featureset $\mathcal{F}$ does not have zero-group-knowledge (using Theorem 6) : $\mathcal{C}_{\mathcal{F}} = \{C_1, C_2, C_3, C_4\}$ with $C_1 = \{x_1, x_5\}$, $C_2 = \{x_2, x_6\}$, $C_3 = \{x_3, x_7\}$, and $C_4 = \{x_4, x_8\}$. Consider the classifier $h \in \mathcal{H}_{\mathcal{F}}$ with $h^{-1}(1) = \{C_1, C_3\}$. Then $L_P^{EO}(h) = \sum_{l \in \{0,1\}} \left| \frac{P(h^{-1}(|1-l|) \cap X_{A,l})}{P(X_{A,l})} - \frac{P(h^{-1}(|1-l|) \cap X_{D,l})}{P(X_{D,l})} \right| = |\frac{1}{2} - 0| + |0 - \frac{1}{2}| = 1$. Thus $U_{adv}^{EO}(\mathcal{F}) = 1$.

- **(demographic parity)** Consider the domain $X = x_1, x_2, x_3, x_4$ with $A = x_1, x_2$ and $D = x_3, x_4$. Furthermore let $\mathcal{F} = \{f_1, f_2\}$ with $f_1^{-1}(1) = \{x_1, x_3\}$ and $f_2^{-1}(1) = \{x_1, x_4\}$. Thus, $\mathcal{C}_{\mathcal{F}} = X$. Furthermore let $P$ be uniform over $X$, i.e. $P(\{x_1\}) = P(\{x_2\}) = P(\{x_3\}) = P(\{x_4\}) = 0.25$. We have demographic parity for both features. However, since $\mathcal{C}_{\mathcal{F}} = X$, the featureset does not have demographic parity. Furthermore, the information from the cells suffice to perfectly predict the group-membership.

- **(calibration parity)** Consider the same domain $X$, the same feature set and the same probability distribution $P$ as in the case of zero-group-knowledge. Furthermore consider the featureset $\mathcal{F} = \{f_1, f_2\}$ with $f_1^{-1}(1) = \{x_1, x_3, x_5, x_7\}$ and $f_2^{-1}(1) = \{x_1, x_4, x_6, x_7\}$. Both features of $\mathcal{F}$ have calibration parity, since both sides of each split have success-rate 0.5 for each group. Furthermore the $\mathcal{F}$ itself does not have calibration parity: We have $\mathcal{C}_{\mathcal{F}} = \{C_1, C_2, C_3, C_4\}$ with $C_1 = \{x_1, x_7\}$, $C_2 = \{x_2, x_8\}$, $C_3 = \{x_3, x_5\}$, and $C_4 = \{x_4, x_6\}$. Both cells $C_1$ and $C_2$ have one element from $X_{A,1}$ and one from $X_{D,0}$. Thus the success rate of elements of group $A$ is 1 in these cells and the success rate of elements of group $D$ is 0. Accordingly, both cells $C_3$ and $C_4$ have one element from $X_{D,1}$ and one from $X_{A,0}$. Thus the success rate of elements of group $A$ is 0 in these cells and the success rate of elements of group $D$ is 1 Thus when splitting these cells by group-membership both cells the resulting scores don't remain the same.

$\square$