# OpenReview forum: "Impossibility results for fair representation"
_NeurIPS.cc/2021/Conference — NeurIPS 2021 Submitted_

### Official Review · Reviewer_58vf · 2021-07-08

**Rating:** 6
**Confidence:** 4

**Summary:**

This paper formalizes impossibility results for fair representation learning. The authors present a series of settings where various desirable properties cannot be achieved.

**Limitations And Societal Impact:**

Yes

**Main Review:**

I think this paper makes a valuable contribution. Some of the results are fairly intuitive, but it's helpful to see them formally specified.

There's a sense in which the results, particularly in Section 4, aren't really about representations per se. It would've been helpful to see a general version of the step from Lemma 1 to Theorem 1 -- something like, if $h_1$ and $h_2$ don't satisfy some property w.r.t. both $P_1$ and $P_2$, there is no representation that can in the adversarial case.

I found the results on "fairness of a feature" to be fairly interesting.

I'd also appreciate a bit of caution in stating these results. They're in some sense worst-case, since they're about the existence of an alternate marginal under which a learned representation would perform poorly. But this doesn't say anything about real-world cases, which may be better than the worst case.

- l. 289: "incompetability" -> "incompatibility"
- l. 350: $\eta$ and $\epsilon$ are reversed


**Time Spent Reviewing:**

2

---

> ### Author Response · Authors · 2021-08-10
> **Initial response**
>
> Regarding your comment about Lemma 1 and Theorem 1: Yes, some more explanation should be in place here we will add them in the final version; To be a good fair representation for a learning problem, the representation should meet two requirements: 1) it allows the expression of a good model for the problem (a low-error classifier for a classification problem), and 2) That every model expressible with this classification is fair (this is for the adversarial agents setup). Therefore, given two tasks P_1, P_2, a representation that is a good fair one for both, should 1) allow expressing classifiers h_1 and h_2 that are low-error for these tasks respectively, and, 2)  due to the second requirement, since both h_1 and h_2 are expressible, they should both be fair w.r.t. both tasks. This is what Lemma 1 rules out. We will be happy to elaborate more if you think it is needed.
>
>
>
>
>
> Regarding your comment about our analysis being worst-case.  When one wishes to provide defence against malicious agents (which is the case when addressing adversarial agents), it is realistic, in fact needed, to take into account every possible action of such an agent. In regards only considering worst-case distribution shifts: Prior works which claim to have found representations that guarantee fairness and are transferable to different tasks. Our work is aimed at showing the limitations of such an approach. While this does not mean that fair representations need to be futile in any application, any user should be aware of these possible limitations (i.e. that no non-trivial representation can guarantee fairness for any distribution shift) and should be alarmed/change their representation/classifier once an adversarial/unfortunate distribution shift happens. Furthermore while our Claim 1 and Claim 2 only state the existence of such a distribution shift, picking a distribution shift which changes the fairness is not limited to artificial examples. In fact, distribution changes which do not change the adversarial fairness of a representation at all seem far less likely in reality.
> Thank you for finding typos! We will fix them in our final version!
> We thank the reviewer for their helpful comments!

---

### Official Review · Reviewer_NE9z · 2021-07-15

**Rating:** 5
**Confidence:** 3

**Summary:**

In this work authors analyzed the transferability of fair representations and showed that it is impossible to train a classifier using fair representations that can achieve the fairness goal for all the cases.

**Limitations And Societal Impact:**

Authors mentioned that they do not see negative societal impacts related to their work and also did not provide sufficient discussion about the limitations of their work. Authors can maybe talk about limitations associated to their narrowed taxonomy.

**Main Review:**

The paper targets an interesting concept that is also timely and new; however, there are some concerns:
1. As authors also point out, having fair representation can mean different things. I do not think that the taxonomy that the authors gave here are accurate or general enough to be able to get to a general conclusion. As authors ground their work based on these definitions, I find it difficult to draw any significant conclusions based on the results.

2. In addition, the paper is written and organized very poorly. Some major proofs are not placed in the main text which make it hard to follow the paper. The paper also has some writing issues as well some pointed below, but the organization is something that needs more work. Some grammar issues:
line 231: is might not sufficient -> is not sufficient. line 382: that claim to learned -> that claim to learn.

3. The paper also jumps from one definition to another without clear reason. I would suggest authors to maybe focus on DP as many work in fair representation learning focuses on DP and put all the major discussions along with the proofs for DP in the main text and put discussion about equalized odds in the appendix. As authors also point out work in DP is more so it would make more sense to put the focus on DP instead of Eqodds.

4. I am also curious to see what authors think about information theoretic work in this area. Maybe some brief discussion should be provided on different types of techniques people use and where they fall under according to their taxonomy. I again generally find their taxonomy narrow and have difficulties about figuring out which work may fall under which section, so maybe some clarity or even changing the taxonomy can be beneficial.



**Time Spent Reviewing:**

3 hours

---

> ### Author Response · Authors · 2021-08-10
> **Initial response to Review NE9z**
>
>
> While, of course, our conclusions apply only to the situations we define in our taxonomy, there is an extensive literature that discusses such situations (as elaborated in our previous work section, both in the submission body and in the appendix). Our results indicate that much of that body of papers, some of which are from renowned authors and are heavily cited, is misguided. In this respect our contributions are clearly significant.
>
>
>  2. “In addition, the paper is written and organized very poorly. Some major proofs are not placed in the main text which make it hard to follow the paper. The paper also has some writing issues as well some pointed below, but the organization is something that needs more work. Some grammar issues: line 231: is might not sufficient -> is not sufficient. line 382: that claim to learned -> that claim to learn.”
> Thanks for finding the typos and grammar mistakes! We will fix them in the final version!
> Proofs are moved to the appendix due to space constraints as is standard. Can you please be more specific about any other way in which you think the paper is poorly organized? In any case, we keep working to improve the presentation.
>
>
> 3)As explained in our introduction, DP is a rather limited notion of fairness. Its main merit is that it is easy to explain and work with, and this probably contributes to its popularity. EqOdds is a rather common fairness notion that was designed to address some of the inherent limitations of DP. Chapter 2 of the book “Fairness in Machine Learning” https://fairmlbook.org/ provides an extensive discussion of this issue.
>
>
>
>
> 4. There are many interesting topics and approaches related to fairness in ML. Clearly one paper cannot cover all of those. We think that a review should address the topics a submission is about. While, of course, our conclusions apply only to the situations we define in our taxonomy, there is an extensive literature that discusses such situations (as elaborated in our previous work section, both in the submission body and in the appendix). Our results indicate that much of that body of papers, some of which are from renowned authors and heavily cited, is misguided. In this respect our contributions are clearly significant.
>
>
>
> Limitations And Societal Impact:
>
> It is not clear to us in what ways you perceive our taxonomy as narrow. As we mentioned above, there is a large body of work to which our results are relevant and significant.
> As for potential negative implications of our work: Our paper aims to call for more caution when applying fairness strategies proposed by previous research. Calling for such critical scrutiny does not carry social risks, but rather mitigates such risks.
>
> We thank the reviewer for their feedback!

---

> > ### Comment · Reviewer_NE9z · 2021-09-02
> > **Response to Authors**
> >
> > I read author responses, but am still concerned about the organization of the paper. Here is my suggestion to make the paper more organized:
> > As also mentioned before, you can either stick to one definition of fairness for the main body, or clearly have devisions in your paper showing when which definition is being addressed. I am well aware of the existing different fairness definitions and what EqOdds is (btw EqOdds also has limitations and it is not the case that DP is limited and not EqOdds). I am just suggesting that since work is more around DP in learning fair representations maybe sticking to DP would be better option; however, you can still talk about both as long as the paper is organized such that the reader does not have to go back and forth for each definition. Also maybe summarizing all the findings for each definition for each taxonomy can be useful as well at the end again in organized subsections.
> >
> > You should also think about how to sell your taxonomy better. Why are these chosen? and what is the significance of these in defining fair representations?
> >
> > Avoiding corss-references to the appendix or other parts of the paper can also be beneficial for the organization of the paper. (I acknowledge the space limitations, but if something is important to be referenced and cut the flow of  the paper maybe it should be incorporated somehow either by brief description about the matter or cutting other unimportant discussion to incorporate the important ones.)
> >
> > Overall, I think the paper has potential, but I still think the paper is slightly below the acceptance threshold due to these organizational issues and issues around the taxonomy. If the paper is written in a more accessible fashion it will be a more promising paper.

---

> ### Author Response · Authors · 2021-09-09
> **The reviewer NE9z addressed only  "organizational issues" and not the results, correctness, potential impact, etc.**
>
> Our paper is quite unusual. We show that a line of work that has been going on since at least 2013 with contributions from world top groups (From Berkeley, U of Toronto, Harvad, CMU, Edinburgh and more) is just futile! Namely,  that there cannot exist any data representation that guarantees fairness over multiple tasks (w.r.t. two most popular notions of fairness)! However, the review says nothing about our results, their correctness, novelty or potential impact. It just complains about "organizational issues" of the paper and some unspecified dissatisfaction with our introductory taxonomy and concludes "reject". How can a potentially impactful work be rejected based on such arguments?
> NeurIPS deserves better -  a more serious reviewing process.

---

> > ### Comment · Reviewer_NE9z · 2021-09-09
> > **Response to Authors About their Concerns**
> >
> > I acknowledged the contributions of the work in my response specifically; however, if authors believe that clarity, coherency and accessibility of their paper is not important in such a top tier conference and are not agreeing with these comments, then the paper should definitely get below threshold score since authors do not want to comply with the comments provided to improve their work further. I again repeat myself the paper has potential, but needs significant improvements! I also outlined specifically what the issue is with the taxonomy, but again it seems like authors do not agree with any suggestion it is made for them to improve their work.

---

> > > ### Author Response · Authors · 2021-09-10
> > > **Reply to Reviewer NE9z**
> > >
> > > We would like to clarify that our disappointment with the review is in no way an indication
> > > that we will not take into consideration their advice and do our best to improve
> > > the writeup.
> > > More concretely, the moving of proofs to the appendix is due to the submission space limitations and we will incorporate
> > > them in the main text in any version that allows more space.
> > > As for the suggestion to focus on the DP notion of fairness, we would like to call the attention of the reviewer
> > > to the fact that this would considerably limit the message of our paper. We have in our submission a strong impossibility
> > > result for Odds Equality as well, and we think it is important to keep it in the paper.  We will however work on making it more clear which notion is used in a given part of the paper, maybe dividing each section to subsections by fairness notions. We thank the reviewer bringing it to our attention that this was an issue.
> > >
> > > We will also make an effort to improve the presentation of our taxonomy as well as add explanations and motivate it further.

---

### Official Review · Reviewer_1FfL · 2021-07-16

**Rating:** 5
**Confidence:** 3

**Summary:**

This paper demonstrates two impossibilities raised in a fair machine learning context: (i) the impossibility of achieving a generic fair data representation; (ii) the impossibility of achieving fairness notions (except disparate parity) in the case of manipulating a single feature. The paper also answers the two questions raised in Creager et. al. [3] via theoretical analysis under binary classification setting w.r.t. varying fairness notions.

**Limitations And Societal Impact:**

A clear discussion of societal impacts is provided in the paper. In spite of the numerous questions raised, however, this paper does not adequately address limitations and previous works, especially regarding multi-task fairness problems. See the 1st point in “Main Review”.

**Main Review:**

The paper raises important and practically relevant questions concerning multiple fairness notions. Nevertheless, several points need to be clarified. First, as demonstrated in 1.1, what is the difference between malicious and fairness-driven agents? The clear distinguishment between the three groups of agents should be stated through rich prior works; this also holds for the criteria of group fairness notions.  There are also significant problems with the presentation of this paper. For example, it is not clear why the authors looked into adversarial fairness (malicious decision-maker) and then multi-task setting afterward. Lastly, while the proof of Claim 1 does not have any significant technical contributions, Claim 2 and Collorary 1 are rather straightforward. Additionally, Theorem 3 only demonstrates a feature set that can lead to the worsening of unfairness; there can be certain cases in which adding or subtracting a feature improves fairness and performance. To conclude, the overall presentation is difficult to follow. Organizations and descriptions of the proofs can be improved. In addition, the authors used the same lines from Zemel et. al. [15].

**Time Spent Reviewing:**

6

---

> ### Author Response · Authors · 2021-08-10
> **Initial response**
>
> We explicitly define our notions of types of agents. Clearly stated definitions are preferable to the vague approach of ‘’stated through rich prior works” that the reviewer seems to suggest. We will also add more explicit references to papers where these notions have some up.
> The optimization criterion for the malicious and the fairness-driven agent are opposed. Given a representation and information about the ground truth a fairness-driven agent will always pick a classifier that best fulfills some fairness and accuracy constraints. The malicious agent in contrast always picks the most unfair classifier. We make this distinction explicit in our writing.
>
> Regarding the comment “For example, it is not clear why the authors looked into adversarial fairness (malicious decision-maker) and then multi-task setting afterward.”
>
> We set up the ground for our results, and that requires explaining the types of agents we address as well as the setup of multi-task that our paper, as well as much of the fair representations literature address. We do not see any issue with this order of definitions.
> The multi-task setting we are looking at is with respect to the notion of adversarial fairness for presentations.
>
>
> Having some claims that can be shown without “technical contributions” is in no way a deficiency of a paper (in particular since the paper has other claims that do require non-trivial technical reasoning). As for Theorem 3, you seem to miss its main point, which is showing that the same feature on the same task can be either beneficial or detrimental to fairness. This phenomenon has not been shown before.
> We agree that the main contribution of our paper is not technically impressive, but it goes against a current trend in this field of research. We believe that this is a valuable contribution to the field as it will redirect research efforts. How technically impressive our proofs are is not substantial for achieving this contribution.
>
> Regarding your comment about using the same lines as Zemel et. al.[15]:
> Please note that the lines from Zemel et al are cited with quotation marks. We clearly state that these lines are from that paper since we wish to contradict what they imply.
>
> We will work to improve the presentation for our final version to make our results more accessible.
> We thank the reviewer for their feedback!

---

### Official Review · Reviewer_qgAQ · 2021-07-17

**Rating:** 6
**Confidence:** 3

**Summary:**

This work establishes a formal framework for exploring impossibility questions related to fair data representations. Impossibility results are proven within this framework with an emphasis on challenging prior work in this area.

**Limitations And Societal Impact:**

The paper gives somewhat of a non-answer to the question of potential negative social impact. I’d like to see more discussion of limitations and potential negative impact such as how this work could potentially be misconstrued in a harmful way.

**Main Review:**

The paper takes an interesting and important stance, critiquing a popular approach to fair classification (fair representations) and exploring its limitations. However, the claims seem a bit exaggerated with respect to prior work in relation to the perspective studied in the paper. The results here are narrowly tailored to a specific setting that assumes objective access to ground truth and appears opposed to the notion of demographic parity. E.g., the paper claims to resolve two open questions from [4], but it only resolves them under certain assumptions. This ground-truth-dependent perspective is useful for theoretical analysis and insights, but is unrealistic and fails to capture the complexity of fairness problems. These issues aside, I still feel the paper makes relevant contributions, including reining in overblown claims in prior work such as the promise of task independence.

As the paper notes, many of the results are straightforward and not necessarily technically complicated or surprising. Nevertheless, a lot of work has gone into defining and understanding a new perspective on an area of research. So the perspective, framework, and results shown are novel and have significance.

The clarity is okay, but could be improved and I found the paper hard to follow at times. The introduction could be condensed and restructured to decrease length and reduce redundancy. This would improve readability/clarity and give more room for exposition elsewhere or more discussion of related work as currently appears in the appendix.

Overall, I would like to see the claims toned down and the limitations of the ground-truth-dependent perspective acknowledged more. The paper makes valuable contributions that are hampered by the writing and limitations of the framework. It is a borderline accept in my opinion.


Additional comments:

The focus on ground truth leads to some questionable implications, which may be due in part to unfortunate phasing. On page 2, for example, assuming correlation between group membership and ground truth as stated in the paper is different from assuming correlation between group membership and labels in a training set. The latter allows for sample bias, label bias, etc. The former is a little too close to assuming inherent differences between the groups.

Likewise, the section on “Viewing perfectly accurate decisions as fair” says, “It makes a lot of sense in tasks like conviction in a crime - if you convict all criminals and no one else, you should not be accused on unfairness.”
Criminal justice is a problematic example here given the high-profile issues with inaccuracy and discrimination in this area that many fairness studies attempt to address. Further, the goals of criminal justice are often one-sided rather than aiming for the impossible goal of perfect accuracy (either minimizing the risk of false conviction at the expense of releasing criminals or alternatively, minimizing the risk of releasing criminals at the expense of false conviction).

Grammar:
“A representation is fairness-driven fair if there exists a loss minimizing (or an approximate minimizer) classifier based on that representation is fair”
“we restrict our discussion the case of one binary protected attribute”
“…this requirement is might not sufficient for guaranteeing that an accuracy232
driven decision maker arrives at a fair decision”
“…since f is not a constant over X there is are points in the other group on which f has the opposite value.”


**Time Spent Reviewing:**

5 hours

---

> ### Author Response · Authors · 2021-08-10
> **Initial response to Reviewer qgAQ**
>
> “The paper takes an interesting and important stance, critiquing a popular approach to fair classification (fair representations) and exploring its limitations. However, the claims seem a bit exaggerated with respect to prior work in relation to the perspective studied in the paper. The results here are narrowly tailored to a specific setting that assumes objective access to ground truth and appears opposed to the notion of demographic parity. E.g., the paper claims to resolve two open questions from [4], but it only resolves them under certain assumptions. This ground-truth-dependent perspective is useful for theoretical analysis and insights, but is unrealistic and fails to capture the complexity of fairness problems. “
>
>
> We use ground truth for the purpose of defining what should be considered fair, not as a tool for solving the problem. In many cases getting correct ground truth labels is realistic (did the student succeed or fail in the program, did the customer return the loan or not). Throughout machine learning ground truth is used to define success, regardless of how hard it may be to discover it.
>
>
>
>
> Yes, we do address tasks in which there exists an inherent difference between groups.  Inherent differences between groups exist for many tasks (we give such an example below) and turning a blind eye to it based on ideology will not help address such differences. We are not claiming that the group memberships is the *cause* of the difference in performance, nor do we *Assume* that the objective truth does not agree with DP. We just acknowledge that such correlations exist and discuss what should be considered fair in such cases.
>
> Consider, for example, success in passing drivers license tests when the groups are defined by age. In my hometown, the success rate among people taking their first drivers test in the group “People  over the age of 50” is only 60% of the success rate in the group “17 year old people”. Does this mean that the test results are unfair? In such a case there are “inherent differences between these groups” when it comes to learning to drive.
> bias in the data: we acknowledge that bias in the data is a real life problem. However, as our results are mainly negative, our message is that even if we were to have access to the ground truth, a representation cannot guarantee fairness for multiple tasks. Thus if the situation is more complex as we acknowledge it is in real life application, this limitation still holds as one cannot hope for constructing a representation that guarantees fairness without full knowledge of the ground truth if one fails even with full knowledge.
>
>
> We are not talking about the goals of the criminal justice system, all we are saying is that our definition is such that if we had an ideal judge that could miraculously find out who is a criminal and who is not, such a judge would not be considered unfair. The questions we address is how can fairness be defined for imperfect decision makers.
>
> Thank you for finding the typos! We will fix these in our final version!
>
> Our paper aims to call for more caution when applying fairness strategies proposed by previous research. Calling for such critical scrutiny does not carry social risks, but rather mitigates such risks.
> We believe that our claims are fully supported by our results. Can you point to any specific claim that we make and you think is not supported by our results?
> We thank the reviewer for their feedback!

---

> > ### Comment · Reviewer_qgAQ · 2021-09-02
> > **Acknowledging author response**
> >
> > I want to acknowledge reading the response and apologize that the authors may gotten the wrong impression from my review. I’ll try to clear things up.
> >
> > There seems to be a concern that I had an ideological disagreement with the notion of inherent differences between groups in certain instances. I do not. What I meant to communicate with my “Additional comments:” is that writing on these sensitive topics should be more careful and thoughtful than it is at times in the paper. This was not a major factor in my evaluation of the research merits, but I do think it’s important.
> >
> > Along the same lines, I’ll encourage the authors again to consider investigating and noting potential negative societal impact. For example, even valid criticism of algorithmic fairness approaches has the potential to be misused if not done carefully.
> >
> > A bigger issue with the writing for me is overall clarity and presentation, especially since the results themselves are not overly complex. I recommend restructuring/rewriting everything before Section 4 to be both clearer and more concise. Then, add in additional content from the appendix.
> >
> > I will also clarify that I was not assuming ground truth was being used as a tool for solving the problem. My concerns about limitations, assumptions, and connections to prior work remain.

---

> > > ### Author Response · Authors · 2021-09-03
> > > **The reviewer seems to accept our rebuttal of their concerns in the original review but does not chnage their evaluation**
> > >
> > > The main reservation the reviewer expressed in their original review were about " the limitations of the ground-truth-dependent perspective".
> > > We argued this issue in our rebuttal and the reviewer's respond seems to essentially accept our argument. However,
> > >  that response concludes with "My concerns about limitations, assumptions, and connections to prior work remain" when no other concerns (other than that dependence on ground truth) about "limitations, assumptions and connection to previsou work" were epressed at any point by this reviewer.

---

### Decision · Program_Chairs · 2021-09-27

**Decision:**

Reject

**Comment:**

Reviewers thought that the paper addresses an important question in Fair-ML (that is, conditions under which fair data representation leads to fair predictions), and offers potentially significant insights. That said, reviewers found the paper challenging to follow and suggested several concrete steps to improve readability and crystalize the core message. In particular, they suggested authors:
(1) elaborate on the conditions under which their results can be applied to real-world datasets (for example, if the so-called ground-truth label in the training data is itself the result of biased decisions and practices, are the results still applicable?
(2) Avoid making blanket statements about highly nuanced and controversial topics, such as fairness of automated decision-support tools in the criminal justice context---as expressed in the passage “if you convict all criminals and no one else, you should not be accused of unfairness.”.
(3) Strengthen the connection between different parts of the paper (e.g., between sections 4 and 5, definitions of adversarial vs. fairness-aware vs. accuracy-aware agents, various fair representation formulations).
(4) acknowledge the worst-case nature of some of the results and their implications (or lack thereof) for real-world settings.
Considering the authors' responses to their suggestions, several reviewers expressed doubt that authors will revise their paper appropriately if accepted. The area chair re-assessed several key points of disagreement between authors and reviewers and concurred with the reviewers that while the paper has the potential for impact, authors need to enhance clarity and accessibility before their work is ready for publication in top-tier venues.